# Tanycytes control the hormonal output of the hypothalamic-pituitary-thyroid axis

Helge Müller-Fielitz[1], Marcus Stahr[1], Mareike Bernau[1], Marius Richter[1], Sebastian Abele[1], Victor Krajka[1], Anika Benzin[1], Jan Wenzel[1], Kathrin Kalies [2], Jens Mittag[3], Heike Heuer[4], Stefan Offermanns[5] & Markus Schwaninger [1]

The hypothalamic–pituitary–thyroid (HPT) axis maintains circulating thyroid hormone levels in a narrow physiological range. As axons containing thyrotropin-releasing hormone (TRH) terminate on hypothalamic tanycytes, these specialized glial cells have been suggested to influence the activity of the HPT axis, but their exact role remained enigmatic. Here, we demonstrate that stimulation of the TRH receptor 1 increases intracellular calcium in tanycytes of the median eminence via $G\alpha_{q/11}$ proteins. Activation of $G\alpha_{q/11}$ pathways increases the size of tanycyte endfeet that shield pituitary vessels and induces the activity of the TRH-degrading ectoenzyme. Both mechanisms may limit the TRH release to the pituitary. Indeed, blocking TRH signaling in tanycytes by deleting $G\alpha_{q/11}$ proteins in vivo enhances the response of the HPT axis to the chemogenetic activation of TRH neurons. In conclusion, we identify new TRH- and $G\alpha_{q/11}$-dependent mechanisms in the median eminence by which tanycytes control the activity of the HPT axis.

[1] Institute for Experimental and Clinical Pharmacology and Toxicology, University of Lübeck, Ratzeburger Allee 160, 23562 Lübeck, Germany. [2] Institute of Anatomy, University of Lübeck, Ratzeburger Allee 160, 23562 Lübeck, Germany. [3] Department of Internal Medicine, Molecular Endocrinology, University of Lübeck, Ratzeburger Allee 160, 23562 Lübeck, Germany. [4] Leibniz Research Institute for Environmental Medicine, Auf'm Hennekamp 50, 40225 Düsseldorf, Germany. [5] Department of Pharmacology, Max-Planck-Institute for Heart and Lung Research, Ludwigstraße 43, 61231 Bad Nauheim, Germany. Correspondence and requests for materials should be addressed to H.M.-F. (email: helge.mueller-fielitz@pharma.uni-luebeck.de) or to M.S. (email: markus.schwaninger@pharma.uni-luebeck.de)

The hypothalamic–pituitary–thyroid (HPT) axis modulates key physiological processes including brain development, cardiovascular function, basal energy metabolism, and the regulation of body temperature[1]. Disturbances of the axis are frequent, affecting almost 6% of the US population and 10.5% of Europeans, and can have a severe impact on health[2, 3]. To ensure a tight control of thyroid hormones, the HPT axis is regulated at multiple points.

Thyrotropin-releasing hormone (TRH) is produced by hypophysiotropic neurons that are located in the paraventricular nucleus (PVN)[4, 5] and project into the median eminence (ME) of the hypothalamus[6]. In the ME, TRH is released into portal blood vessels and stimulates thyroid-stimulating hormone (TSH) secretion from the pituitary. As a circumventricular organ, the ME is characterized by an open blood–brain barrier that serves as an interface between the neural and peripheral endocrine systems. In the ME a specialized cell population, the tanycytes, has been suggested to control the crossing of blood-borne substances from the periphery into the brain[7] and to function as chemosensors[8–10]. The cell bodies of tanycytes, which are connected by tight junctions, are located in the ependymal layer of the 3rd ventricle and contact the cerebrospinal fluid[7]. Tanycytes send long processes into the parenchyma and are classified as α- and β-subtypes according to their location in the wall of the 3rd ventricle and the direction of their projections[11]. While α-tanycytes reside dorsally, β-tanycytes occupy the ventral sidewall of the 3rd ventricle and line the floor of the 3rd ventricle in the ME[11]. Processes of the latter reach to the portal blood

vessels in the ME, where their perivascular endfeet are closely associated with axon terminals containing releasing hormones, such as TRH[12]. TRH-containing neurons even form synaptoid contacts on tanycyte processes[13].

The strategic localization of tanycytes has suggested their involvement in neuroendocrine circuits. Indeed, tanycytes express the thyroid hormone transporters monocarboxylate transporter 8 (Mct8, Slc16a2) and organic anion transporting polypeptide family member 1C1 (Oatp1c1, Slco1c1)[14] as well as iodothyronine deiodinase type 2[15] (Dio2) that converts thyroxine (T4) into 3,3′,5-triiodothyronine (T3). In addition, tanycytes express the highly specific TRH-degrading ectoenzyme (Trhde, pyroglutamyl peptidase II)[12]. However, whether tanycytes regulate the HPT axis in vivo is unknown.

Here we report that tanycytes control the HPT axis in vivo. Activation of the TRH receptor 1 (TRHR1) elevates the intracellular calcium concentration ($[Ca^{2+}]_i$) in β-tanycytes of the ME through a $G\alpha_{q/11}$-coupled pathway. $G\alpha_{q/11}$ signaling in tanycytes leads to an increase in the size of tanycyte endfeet and an upregulation of TRH-DE that might block hormone secretion into the portal blood vessels. Interrupting the tanycytic response to TRH by deleting $G\alpha_{q/11}$ proteins specifically in tanycytes of mice ($G\alpha_{q/11}^{tanKO}$) reduces the activity of the hypothalamic TRH-DE as well as the TRH receptor-mediated change in tanycyte endfoot size. In the absence of TRH signaling in tanycytes, TSH release is increased after activating TRH neurons. Overall, our data demonstrate an important role of tanycytes in the fast regulation of the HPT axis.

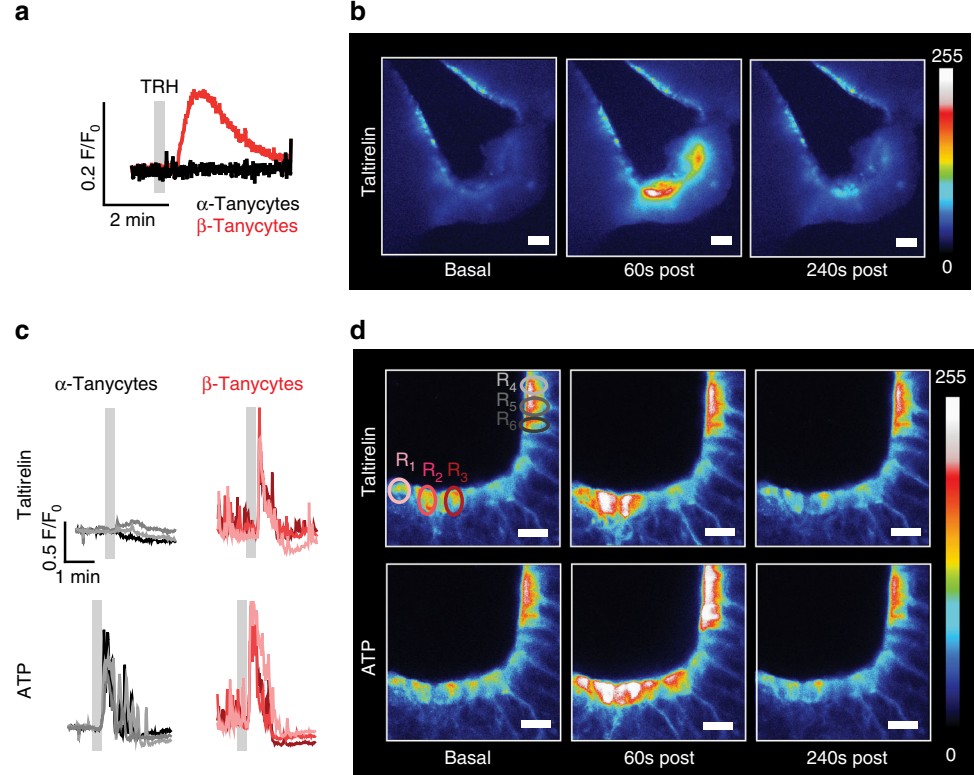

**Fig. 1** TRH receptor agonists increase $[Ca^{2+}]_i$ selectively in tanycytes of the median eminence. **a** and **b** TRH and the TRH receptor agonist taltirelin increased $[Ca^{2+}]_i$ in tanycytes of the ME 2 weeks after rAAV-mediated transduction of the calcium sensor GCamP6s. **a** After treating hypothalamic slices with TRH (33 µM, *gray bar*), only β-tanycytes (*red line*) responded with an increase in $[Ca^{2+}]_i$ but not α-tanycytes (*black line*). Representative traces are shown. **b** Heat maps of fluorescence intensity visualized the location of responsive cells after taltirelin stimulation (33 µM). *Scale bar*, 50 µm. **c** Single-cell calcium responses after stimulation with taltirelin (150 µM, *gray bar, top*) and ATP (150 µM, *gray bar, bottom*) in tanycytes of the ME (*red*, $R_{1-3}$) and the ventricular wall (*gray*, $R_{4-6}$). The regions of interest that were imaged by confocal microscopy are indicated in **d** ($R_{1-6}$). **d** Representative heat maps after stimulation with taltirelin (150 µM, *top*) and ATP (150 µM, *bottom*) in acute brain slices of GCamP6s-transduced mice. Regions of interest are indicated in the ME ($R_{1-3}$, *red*) and the ependymal layer of the 3rd ventricle ($R_{4-6}$; *gray*). *Scale bar*, 25 µm

## Results

**TRH receptor agonists stimulate $[Ca^{2+}]_i$ in β-tanycytes.** To target tanycytes, we injected DNA vectors based on recombinant adeno associated virus (rAAV) 1/2 into the lateral ventricle of mice. The serotypes 1 and 2 are known to hardly diffuse into the parenchyma[16, 17]. After injecting the rAAV vector expressing the Cre recombinase into the lateral ventricle of Ai14 reporter mice, we observed expression of the reporter tdTomato in the ependymal layer suggesting that the rAAV1/2-based vector was trapped in the ependymal layer of the ventricular system (Supplementary Fig. 1a–d).

Using this approach we measured $[Ca^{2+}]_i$ in acute brain slices. We expressed the calcium sensor GCaMP6s[18] in tanycytes by injecting AAV-CAG-GCaMP6s into the lateral ventricle of mice. Stimulation of acute brain slices with TRH selectively increased $[Ca^{2+}]_i$ in β-tanycytes of the ME but not in α-tanycytes (Fig. 1a). Taltirelin, a TRH receptor agonist, had the same effect (Fig. 1b; Supplementary Movie 1). Imaging by confocal microscopy showed that in response to taltirelin $[Ca^{2+}]_i$ rose in cell bodies and in the projections of individual β-tanycytes but not in α-tanycytes (Fig. 1c, d). In contrast, ATP elevated $[Ca^{2+}]_i$ in both α- and β-tanycytes (Fig. 1c, d).

Cotreatment with the TRH receptor antagonist midazolam[19] inhibited the $[Ca^{2+}]_i$ increase after activation of TRH receptors in a reversible manner (Fig. 2a; Supplementary Movies 2, 3) but had no effect on the ATP-induced response (Fig. 2b). Because taltirelin and midazolam act on both TRHR1 and TRH receptor 2 (TRHR2)[20], we investigated knockout mouse lines for both subtypes. In $Trhr1^{-/-}$ mice taltirelin lost its stimulatory effect on $[Ca^{2+}]_i$ (Fig. 2a, c), whereas ATP as positive control[8, 9] was still effective in increasing $[Ca^{2+}]_i$ in α- and β-tanycytes (Fig. 2b, c). In contrast, in $Trhr2^{-/-}$ mice taltirelin stimulated $[Ca^{2+}]_i$ increase as in wild-type animals (Fig. 2a). After laser microdissection of α- and β-tanycytes, we detected $Trhr1$ messenger RNA (mRNA) in β-tanycytes but not in α-tanycytes (Supplementary Fig. 2) in accordance with in situ hybridization data of the Allan Brain Institute (experiment: 79591633)[21].

TRHR1 is coupled to $G\alpha_{q/11}$ proteins and activates phospholipase C (PLC) and inositol-1,4,5-trisphosphate ($IP_3$) production[22]. To investigate the involvement of this pathway in the increase of $[Ca^{2+}]_i$ we deleted $G\alpha_q$ and $G\alpha_{11}$ proteins in glial cells and tanycytes with the help of the Cre-driver mouse line $GlastCreER^{T2}$ ($G\alpha_{q/11}^{gliaKO}$ mice; for an overview about mouse models, see Supplementary Table 1). Double deficiency of $G\alpha_q$ and $G\alpha_{11}$ is usually required to investigate the physiological role of these two closely related G proteins as they are able to compensate for each other[23]. In $G\alpha_{q/11}^{gliaKO}$ mice, the $[Ca^{2+}]_i$ response after stimulation of the TRH receptors was reduced (Fig. 2d). Moreover, pretreatment of acute brain slices with the PLC inhibitor U73122 or with the $IP_3$-receptor antagonist 2-APB blocked the taltirelin effect (Fig. 2d–f). Thus, TRH and its analog taltirelin are able to stimulate β-tanycytes through TRHR1- and $G\alpha_{q/11}$-mediated signal transduction.

**$G\alpha_{q/11}$ signaling modulates the morphology of tanycytes.** To evaluate the cellular consequences of $G\alpha_{q/11}$ signaling, we adopted a technique that allows both to chemogenetically activate $G\alpha_{q/11}$ signaling and to image tanycytes. For this purpose, we expressed a fusion protein of the mutant human muscarinergic receptor 3 (hM3D) and mCherry[24] in tanycytes. Our previous data confirmed that rAAV 1/2-based vectors get trapped in the ventricle wall (Supplementary Fig. 1). To further enhance selectivity and to limit expression in the ventricle wall to tanycytes, we put the vector under control of the astrocyte and tanycyte-specific $GlastCreER^{T2}$ driver gene[25, 26]. Therefore, we injected the

Cre-dependent AAV-CAG-flex(hM3D-mCherry) vector into the lateral ventricle of $GlastCreER^{T2}$ mice (gTan³ᴰ). After inducing recombination by injecting tamoxifen, hM3D-mCherry expression could be detected in tanycytes of the hypothalamus but in hardly any other brain area (Fig. 3a, b; Supplementary Fig. 3; Supplementary Movie 4). This technique allowed us to visualize the processes and endfoot structure of tanycytes in the ME by high-magnification confocal microscopy (Fig. 3b). Costainings of collagen IV as a vessel marker (Fig. 3a, b), of aquaporin 4 as an astrocytic marker (Fig. 3c), and of neurofilament 200 as a neuronal marker (Fig. 3d) clearly showed that tanycytes in the ventricular wall send processes through the ME and terminate on blood vessels on the ventral surface of the ME (Fig. 3c–e). There was no overlap of the transduced cells with astrocytes or neurons in the ME (Fig. 3c, d).

hM3D can be activated by clozapine-N-oxide (CNO) treatment and is coupled to $G\alpha_{q/11}$ signaling[24]. After loading acute brain slices with Fura-2 to measure tanycytic $[Ca^{2+}]_i$, we confirmed that activation of the fusion protein hM3D-mCherry by CNO and subsequently of $G\alpha_{q/11}$ signaling enhanced $[Ca^{2+}]_i$ in tanycytes of gTan³ᴰ mice (Fig. 3f). As expected, CNO had no effect on tanycytes that did not express hM3D (gTan^Con; Fig. 3f). When we activated $G\alpha_{q/11}$ signaling in tanycytes by treating gTan³ᴰ mice with CNO, the diameter of tanycytic endfeet was increased (Fig. 3g–i).

**Tanycytic $G\alpha_{q/11}$ knockout mice.** To explore the in vivo consequences of the $G\alpha_{q/11}$-mediated increase in endfoot size, we aimed to delete $G\alpha_{q/11}$ proteins selectively in tanycytes. For a conditional knockout approach, the $GlastCreER^{T2}$ mouse line would not have been suitable because of its Cre activity in astrocytes and tanycytes. Instead, we used the promoter of the $Dio2$ gene that is mainly expressed in tanycytes[27] to drive expression from a rAAV-based vector. After injecting AAV-Dio2-iCre-2A-GFP into Ai14 reporter mice, tdTomato was expressed in cells lining the ventral 3rd ventricle and extending processes into the parenchyma, the typical morphology of tanycytes (Fig. 4a). Costainings revealed that tdTomato-positive cells expressed vimentin, a tanycytic marker[28] and contained the Cre recombinase in the nucleus (Fig. 4b). Apart from tanycytes in the ventral wall of the 3rd ventricle, tdTomato-positive cells were also detectable in the subfornical organ and subcommissural organ (SCO, Supplementary Fig. 4a, e). Only a small number of transduced cells were found in the walls of the lateral or the dorsal 3rd ventricle and the choroid plexus, when we used the tanycyte-specific vector AAV-Dio2-iCre-2A-GFP (Supplementary Fig. 4b, c). When we injected AAV-Dio2-iCre-2A-GFP together with AAV-CAG-flex(hM3D-mCherry) as reporter and counted the cells showing signs of recombination, we found that mainly tanycytes were affected (Supplementary Fig. 4f and g). Thus, injecting AAV-Dio2 based vectors into the lateral ventricle allowed us to manipulate tanycytes in vivo.

By injecting AAV-Dio2-iCre-2A-GFP into the lateral ventricle of $G\alpha_q^{fl/fl}::G\alpha_{11}^{-/-}$ mice, we deleted $G\alpha_{q/11}$ proteins in tanycytes (Fig. 4c; $G\alpha_{q/11}^{tanKO}$). As controls we used littermates ($G\alpha_q^{fl/fl}::G\alpha_{11}^{+/-}$) that received AAV-Dio2-GFP ($G\alpha_{q/11}^{Con}$). Four weeks after injecting the vectors, plasma concentrations of TSH, T4, and T3 were determined and revealed no changes in $G\alpha_{q/11}^{tanKO}$ compared to $G\alpha_{q/11}^{Con}$ mice (Fig. 4d–f). Tanycytes express TRH-DE that degrades TRH[12]. When TRH signaling was blocked in tanycytes of $G\alpha_{q/11}^{tanKO}$ mice, TRH-DE activity was decreased in the mediobasal hypothalamus (Fig. 4g). Down regulation of TRH-DE would impair the degradation of TRH in the ME. However, basal TSH plasma levels were normal in $G\alpha_{q/11}^{tanKO}$ mice suggesting that a chronic deficit in TRH degradation by

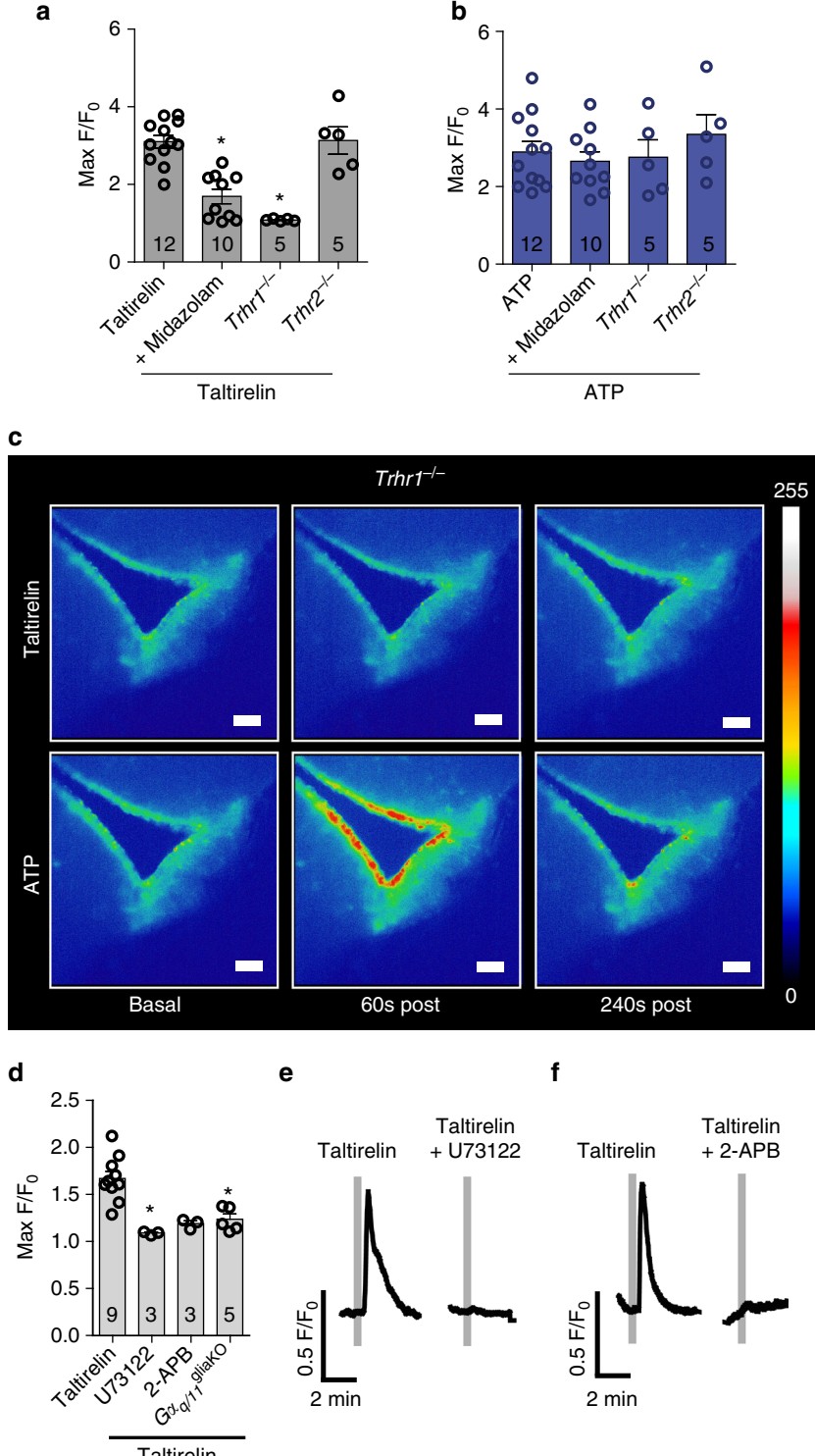

**Fig. 2** TRHR1 activates the $G\alpha_{q/11}$ pathway selectively in tanycytes of the median eminence. **a** and **b** Quantification of taltirelin- (33 µM; **a**) and ATP- (33 µM, **b**) induced maximal $[Ca^{2+}]_i$ response (max $F/F_0$) after treatment with the TRHR antagonist midazolam (500 µM) and in $Trhr1^{-/-}$ and $Trhr2^{-/-}$ mice. $n$, as indicated; *$p < 0.05$ vs. taltirelin stimulation (Kruskal–Wallis with post hoc Dunn's test); mean ± S.E.M. **c** Representative heat maps after stimulation with taltirelin (33 µM, *top*) and ATP (33 µM, *bottom*) in acute brain slices of GCaMP6s-transduced $Trhr1^{-/-}$ mice. *Scale bar*, 50 µm. **d** Quantification of taltirelin-induced maximal $[Ca^{2+}]_i$ response under pharmacological and genetic inhibition of the $G\alpha_{q/11}$ pathway ($G\alpha_{q/11}^{gliaKO}$). $n$, as indicated; *$p < 0.05$ vs. taltirelin stimulation (Kruskal–Wallis with post hoc Dunn's test); mean ± S.E.M. **e** and **f** Representative $[Ca^{2+}]_i$ responses upon taltirelin stimulation (33 µM, *gray bar*) before (*left curve*) and after (*right curve*) treatment with the PLC blocker U73122 (100 µM; 30 min preincubation; **e**) or the IP3-receptor blocker 2-APB (100 µM; 30 min preincubation; **f**)

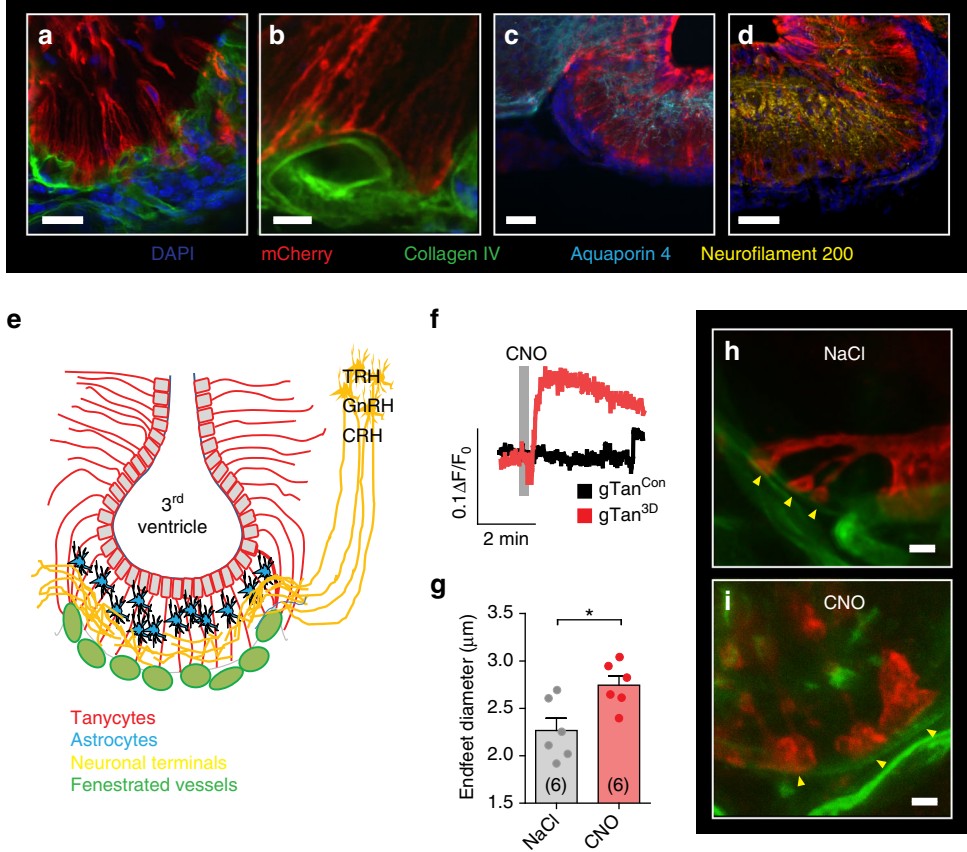

**Fig. 3** Morphology of tanycyte endfeet is modulated by the $G\alpha_{q/11}$ pathway. **a–e** Characterization of the ME structure in coronary sections. Tanycytes were labeled with the membrane-bound protein hM3D-mCherry by injecting AAV-CAG-flex(hM3D-mCherry) in the lateral ventricle of *GlastCreER*[T2] mice and inducing the Cre recombinase with tamoxifen (gTam[3D]). Beta-tanycytes have projections that are branched in their terminal third. The endings of the branches, the so-called endfeet, terminate on the surface of the fenestrated vessels of the portal venous system, located at the bottom of the ME (**a**; *red*, tanycytes, hM3D-mCherry; *green*, vessels, collagen IV). *Scale bar*, 25 µm. **b** High-magnification image of tanycytic endfeet in gTam[3D] mice. *Scale bar*, 5 µm. **c** Astrocytes are present in a layer beneath the tanycytic cell bodies (*cyan*, astrocytes, aquaporin 4). *Scale bar*, 50 µm. **d** Neuronal terminals are located in the central part of the ME (*yellow*, neuronal terminals, neurofilament 200). *Scale bar*, 50 µm. **e** Schematic drawing of the ME structure. **f** CNO (10 µM, *gray bar*) increased tanycytic $[Ca^{2+}]_i$ in acute brain slices of gTan[3D] mice (*red*) in comparison to untransduced control mice (gTan[Con], *black*). Slices were loaded with the calcium responsive dye Fura-2. Representative traces are shown. **g** Quantitative analysis of tanycytic endfoot diameters in gTan[3D] mice 1 h after injecting 0.9% saline (*gray*; NaCl; i.p.) or CNO (*red*, i.p.). Mean endfoot diameters were determined from 15 to 32 endfeet per animal. *$p = 0.0147$ (two-tailed Student's *t*-test); mean ± S.E.M.; *n*, number of animals as indicated. **h** and **i** Representative images of tanycytic endfeet (*arrowheads*) in gTan[3D] mice 1 h after injecting the vehicle (**h**, 0.9% NaCl) or CNO (**i**). *Scale bar*, 2 µm

tanycytes can be compensated by multiple feedback mechanisms in the HPT axis.

Acutely stimulating TRH receptors by treating control mice with taltirelin enhanced the enzymatic activity of TRH-DE in the mediobasal hypothalamus (Supplementary Fig. 5a). Furthermore, when we labeled tanycytes with hM3D-mCherry by injecting AAV-Dio2-iCre-2A-GFP plus AAV-CAG-flex(hM3D-mCherry) into the lateral ventricle of C57Bl/6 mice (dTan[3D]-Bl6, Fig. 5), taltirelin administration increased the endfoot size very much as activation of $G\alpha_{q/11}$ signaling by CNO did (Figs. 3g–i, 5). However, endfoot expansion in response to taltirelin was lost, when $G\alpha_{q/11}$ signaling was interrupted in tanycytes (dTan[3D]-$G\alpha_{q/11}$[tanKO] Fig. 5) confirming that TRH receptor activation and $G\alpha_{q/11}$ signaling increase the size of tanycyte endfeet. These data suggest that tanycytic degradation of TRH and blocking of its release from the ME by endfoot expansion could be relevant during acute hypothalamic stimulation of the HPT axis.

**Chemogenetic stimulation of TRH release.** To investigate the consequences of an acute endogenous TRH pulse, we aimed at a selective chemogenetic stimulation of hypophysiotropic TRH

neurons that terminate on tanycytes[13]. For this purpose, we injected the vector AAV-TRH-hM3D-mCherry, in which the expression of the fusion protein hM3D-mCherry was under transcriptional control of the rat TRH promoter[29], bilaterally into the PVN of C57Bl/6 mice (PVN[3D]-Bl6). After 2 weeks, hM3D-mCherry could be detected in *Trh* mRNA expressing neurons of the PVN (Fig. 6a–d). Stimulating hM3D by CNO increased cFOS expression in the PVN (Supplementary Fig. 6) confirming the activation of TRH neurons. In parallel, plasma concentrations of TSH were elevated (Fig. 6e). As expected, activation of TRH neurons did not elevate TSH levels in *Trhr1*[−/−] mice (Fig. 6e). Chemogenetic activation of TRH neurons by CNO administration also enhanced pituitary *Fos* mRNA expression in wild-type but not in *Trhr1*[−/−] mice (Fig. 6f).

While most TRH neurons were expressing hM3D-mCherry, some transduced cells were not positive for *Trh* mRNA (Fig. 6a–d). This may be due to an incomplete labeling of TRH neurons by in situ hybridization or off-target activity of the TRH promoter fragment in non-TRH neurons. To evaluate possible off-target effects of our chemogenetic strategy on other neuroendocrine axes mediated by the PVN, we analyzed the

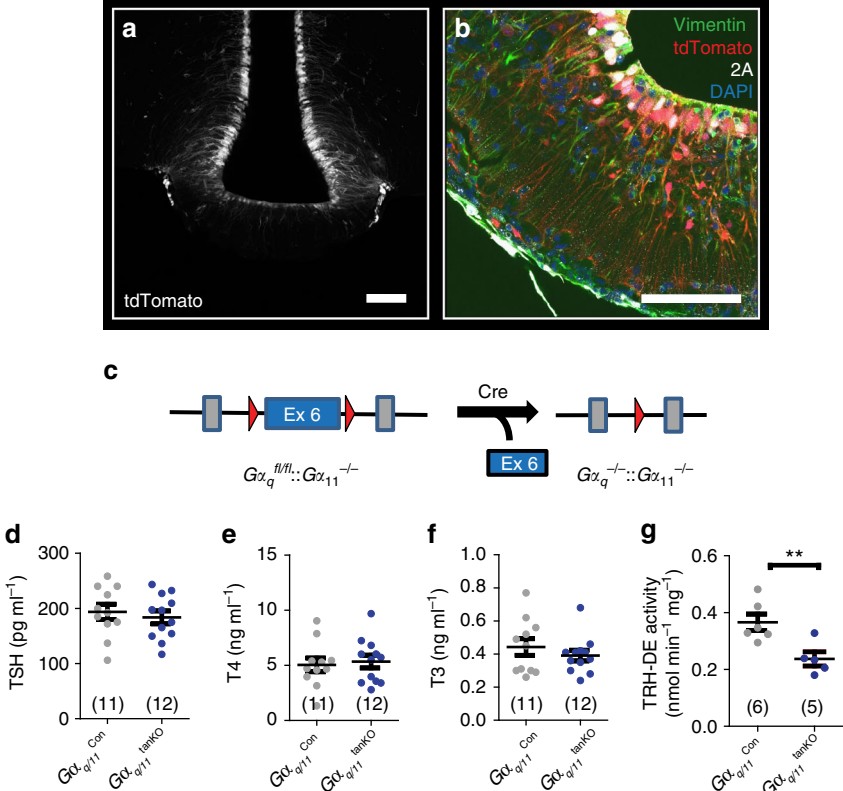

**Fig. 4** Basal activity of the HPT axis is not changed in mice with a tanycyte-specific deficiency of $G\alpha_{q/11}$. **a** tdTomato (*gray*) was expressed in tanycytes 2 weeks after injecting AAV-Dio2-iCre-2A-GFP in Ai14 reporter mice (*scale bar*, 100 μm). **b** Immunofluorescence staining of iCre-2A (*white*; 2A) and vimentin (*green*) showed colocalization in tdTomato-positive tanycytes (*red*; *scale bar*, 75 μm). **c** Knockout strategy to generate $G\alpha_{q/11}^{tanKO}$ mice. Cre recombinase was transduced by injecting AAV-Dio2-iCre-2A-GFP into the lateral ventricle of $G\alpha_q^{fl/fl}::G\alpha_{11}^{-/-}$ mice. AAV-Dio2-GFP injected $G\alpha_q^{fl/fl}::G\alpha_{11}^{-/+}$ into the lateral ventricle were used as control ($G\alpha_{q/11}^{Con}$) **d–f** Basal plasma concentrations of TSH (**d**; $p = 0.59$), T4 (**e**; $p = 0.75$), and T3 (**f**; $p = 0.34$) 3 weeks after inducing the knockout by injecting rAAV ($G\alpha_{q/11}^{Con}$, *gray*; $G\alpha_{q/11}^{tanKO}$, *blue*). **g** TRH-DE activity in $G\alpha_{q/11}^{Con}$ mice (*gray*) and $G\alpha_{q/11}^{tanKO}$ (*blue*) 3 weeks after rAAV injection. **p = 0.0093 (two-tailed Student's *t*-test); mean ± S.E.M.; *n*, as indicated

localization of the transduced hM3D-mCherry-positive fibers in the ME. hM3D-mCherry-positive terminals overlapped with preproTRH (ppTRH; Fig. 6g) and corticotropin-releasing hormone (CRH; Fig. 6h) in the ventral layer of the ME. This is in accordance with reports showing TRH- and CRH-positive fibers in close proximity to the fenestrated capillaries of the portal venous system[12, 30]. However, CNO administration only elevated TSH (Fig. 6i) but not adrenocorticotropic hormone (ACTH, Fig. 6j) plasma concentrations after 2 h. As described previously vasopressin-[31], oxytocin-, and gonadotropin-releasing hormone (GnRH)-positive[32] fibers terminated in a more dorsal part of the ME and were not colocalized with hM3D-mCherry-positive fibers (Fig. 6k–m).

**Tanycytic $G\alpha_{q/11}$ knockout influences the HPT axis activity.** To test whether an acute stimulation of the HPT axis is influenced by the tanycyte-specific deletion of $G\alpha_{q/11}$ signaling, we activated hypophysiotropic TRH neurons in mice with tanycytic $G\alpha_{q/11}$ knockout. For this purpose, we transduced the PVN of $G\alpha_{q/11}^{tanKO}$ and $G\alpha_{q/11}^{Con}$ animals with AAV-TRH-hM3D-mCherry to generate $PVN^{3D}$-$G\alpha_{q/11}^{tanKO}$ and $PVN^{3D}$-$G\alpha_{q/11}^{Con}$ mice. $G\alpha_{q/11}^{Con}$ mice without transduction of the PVN were treated with CNO as control group ($PVN^{Con}$-$G\alpha_{q/11}^{Con}$; Fig. 7a). As shown above (Fig. 6i), activation of TRH neurons by administering CNO to $PVN^{3D}$-$G\alpha_{q/11}^{Con}$ mice increased TSH plasma levels compared to CNO-treated $PVN^{Con}$-$G\alpha_{q/11}^{Con}$ mice (Fig. 7b). A further rise in plasma TSH was observed after activation of TRH neurons in $PVN^{3D}$-$G\alpha_{q/11}^{tanKO}$ mice, in which

TRH signaling in tanycytes was blocked (Fig. 7b). In parallel, we found slightly higher T4 and T3 plasma concentrations in $PVN^{3D}$-$G\alpha_{q/11}^{tanKO}$ animals than in CNO-treated $PVN^{Con}$-$G\alpha_{q/11}^{Con}$ mice (Fig. 7c, d). Activation of TRH neurons also resulted in elevated pituitary *Tshb* mRNA expression in $PVN^{3D}$-$G\alpha_{q/11}^{tanKO}$ mice (Fig. 7e). Thus, acute activation of TRH neurons was only effective on T4 plasma levels and *Tshb* mRNA expression in the pituitary, when tanycyte signaling was inhibited. To evaluate whether the increase in circulating TSH protein is accompanied by a rise in hormone activity, we measured TSH activity by determining cAMP production in JP26-CHO cells stably expressing TSH receptors[33]. When stimulating TRH neurons, the absence of TRH signaling in tanycytes ($PVN^{3D}$-$G\alpha_{q/11}^{tanKO}$ mice) significantly increased plasma TSH activity in comparison to animals with intact tancyte signaling ($PVN^{3D}$-$G\alpha_{q/11}^{Con}$ mice, Fig. 7f). The TSH plasma concentrations and activities correlated (Fig. 7g). Importantly, activation of TRH neurons had a more pronounced effect on the expression of the T3-dependent genes *Thrsp* and *Fasn* in the liver, when TRH signaling in tanycytes was blocked ($PVN^{3D}$-$G\alpha_{q/11}^{tanKO}$; Fig. 7h, i) than under control conditions ($PVN^{3D}$-$G\alpha_{q/11}^{Con}$), supporting the metabolic significance of tanycyte function.

To evaluate whether the tanycytic defect of $G\alpha_{q/11}$ signaling would affect the pituitary response to TRH stimulation, we treated $G\alpha_{q/11}^{tanKO}$ mice with the TRH analog taltirelin and measured TSH plasma levels. TSH plasma levels were not different between mice with and without $G\alpha_{q/11}$ proteins in tanycytes (Supplementary Fig. 5b) arguing against an altered

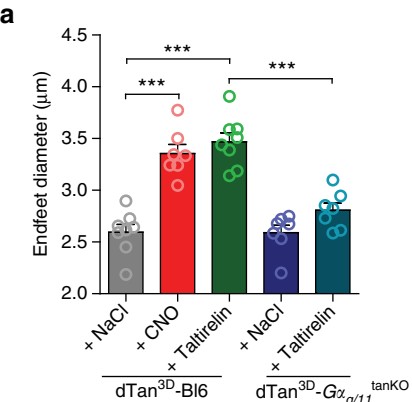

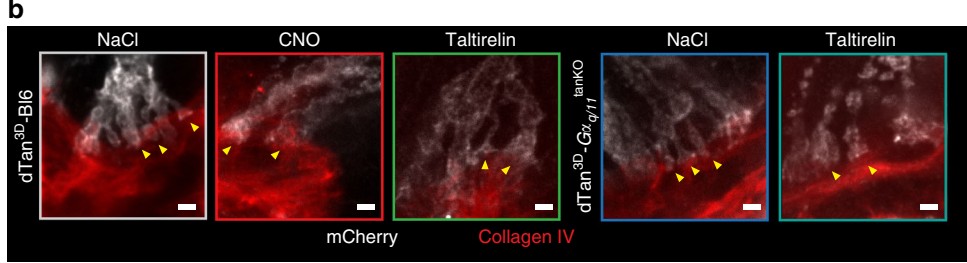

**Fig. 5** Activation of the $G\alpha_{q/11}$ pathway induces changes in tanycytic endfoot diameters. **a** To express hM3D-mCherry in tanycytes, we injected AAV-CAG-flex(hM3D-mCherry) plus AAV-Dio2-iCre-2A-GFP in the LV of C57Bl/6 (dTan3D-Bl6) or $Ga_q{}^{fl/fl}::Ga_{11}{}^{-/-}$ (dTan3D-$Ga_{q/11}{}^{tanKO}$) mice ($2 \times 10^{10}$ genomic particles of each vector in a total volume of 2.5 µl). One hour after treating mice with 0.9% NaCl (dTan3D-Bl6, $n = 8$; dTan3D-$Ga_{q/11}{}^{tanKO}$, $n = 7$), CNO ($n = 7$) or taltirelin (dTan3D-Bl6, $n = 8$; dTan3D-$Ga_{q/11}{}^{tanKO}$, $n = 7$), the diameters of 18–26 tanycytic endfeet were quantified. Mean diameters per animal are presented. One-way ANOVA, $F(4/32) = 29.47$, $p < 0.0001$; ***$p < 0.0001$ between indicated groups (Bonferroni post test); mean ± S.E.M. **b** Representative images of tanycytic endfoot morphology (*yellow arrowheads*) for all groups (mCherry, *gray*; collagen IV, *red*). Scale bar, 2 µm

TRH responsiveness of the pituitary. Thus, TRH signaling in tanycytes inhibits the HPT axis, probably at the level of TRH release from the ME.

## Discussion

In the HPT axis, negative feedback mechanisms are important to stabilize thyroid hormone concentrations within the physiological range. According to previous concepts, thyroid hormones and TSH exert a negative feedback at the level of the PVN[34] and the pituitary[35] by inhibiting the expression of *Trh* and *Tshb*. Tanycytes in the ME now emerge as new players in the fast regulation of the HPT axis. They represent specialized glial cells in the ependymal layer of the medial basal hypothalamus[36] and in other circumventricular organs[28]. Hypothalamic tanycytes can be divided in subgroups: α-tanycytes, which project into hypothalamic nuclei, and β-tanycytes, which project into the ME[36]. The projections of both subtypes end mainly on blood vessels[37].

Recent evidence suggests that tanycytes play a role as biosensors for the fast detection of hormones and transmitters. It has been shown that tanycytes respond with a fast increase in $[Ca^{2+}]_i$ to ATP, glucose[8, 9], histamine, acetylcholine[8], and sweeteners[10]. So far, no other stimuli have been reported to increase $[Ca^{2+}]_i$ in tanycytes. Here, we show that TRH and the TRH receptor agonist taltirelin selectively increase $[Ca^{2+}]_i$ in β-tanycytes of the ME. This effect is mediated by TRHR1, as we detected *Trhr1* mRNA expression in β-tanycytes of the ME and the response was lost in *Trhr1*$^{-/-}$ but not in *Trhr2*$^{-/-}$ mice. TRHR1 is known to signal through a $G\alpha_{q/11}$-coupled pathway[22]. Blocking this pathway either by pharmacological tools or by deleting $G\alpha_{q/11}$ proteins inhibited the elevation of $[Ca^{2+}]_i$ in response to TRHR1 stimulation. These experiments revealed a selective and fast response of β-tanycytes to TRH and led to the hypothesis that tanycytes could modulate the HPT axis via a

TRH-mediated mechanism that controls the release of TRH from the hypothalamus.

To test this concept in vivo, we had to establish a tool that would allow selective activation of hypophysiotropic TRH neurons in the PVN of the hypothalamus. By activating the HPT axis at the level of the PVN, we avoided pituitary effects of exogenous TRH agonists that could override hypothalamic mechanisms. For chemogenetic activation of TRH neurons, we expressed the hM3D receptor fused to mCherry[24] in the PVN under control of the TRH promoter[29]. In situ hybridization identified transduced hM3D-mCherry-positive cells as TRH neurons. Projections of the transduced neurons were visible as hM3D-mCherry-positive fibers in the ventral part of the ME where TRH-[6] and CRH-positive fibers terminate. The specific activation of TRH neurons is supported by the observation that CNO treatment elevated TSH plasma levels but had no effect on ACTH concentrations. Activation of TRH neurons also led to a fast TRHR1-dependent *Fos* mRNA expression in the pituitary, as has been observed after TRH stimulation in vivo[38].

To investigate the physiological function of tanycyte activation in vivo, we developed techniques that allowed for a selective and efficient genetic manipulation of tanycytes. Our approach of targeting tanycytes relied on two factors. First, after intraventricular injection, rAAV1/2-based vectors were trapped in the ventricle wall and did not reach the parenchyma to an appreciable extent. Second, we controlled expression by regulatory gene sequences of *Dio2* or *Glast* that show selectivity for tanycytes and astrocytes. However, due to the administration route and serotype of vectors, only tanycytes in the ventricle wall but hardly any astrocytes in the parenchyma were transduced. For *Dio2*, we identified a short promoter fragment that mediated tanycyte specificity when inserted in rAAV vectors. Alternatively, we used the Cre-driver *GlastCreER*$^{T2}$ mice[25] together with a rAAV-based Flex system[24, 39].

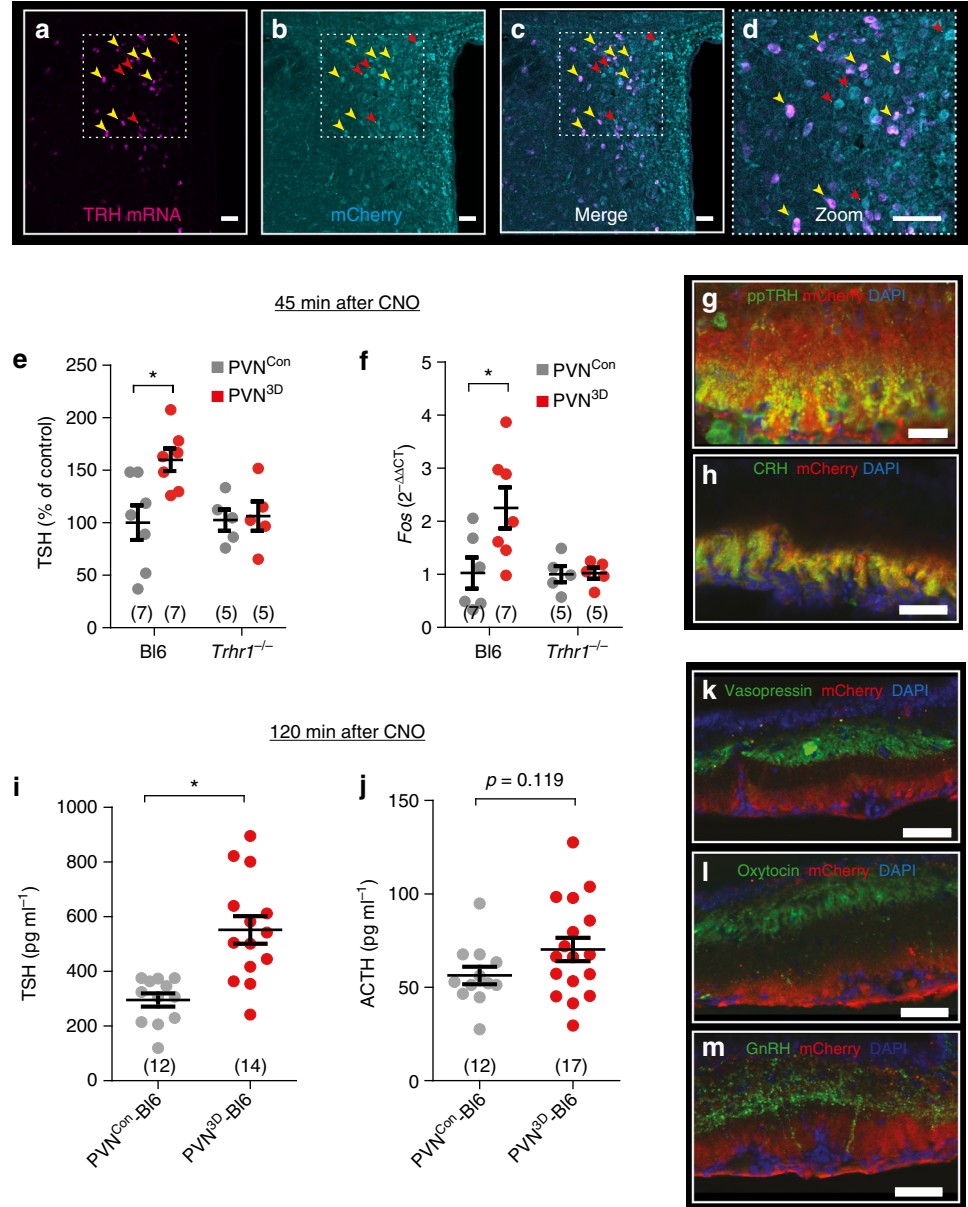

**Fig. 6** Chemogenetic activation of hypophysiotropic TRH neurons increases TSH release. **a–c** Colocalization of *Trh* mRNA (**a**, *magenta*) and hM3D-mCherry (**b**, *cyan*) in the PVN 2 weeks after bilateral injection of AAV-TRH-hM3D-mCherry (PVN^3D). Cells expressing *Trh* mRNA were positive for hM3D-mCherry (*yellow arrowheads*; **c** and **d**). *Red arrowheads* label hM3D-mCherry-positive cells without *Trh* mRNA expression; scale bar, 50 μm. **e** Plasma TSH concentrations 45 min after CNO administration to *Trhr1*^+/+ (Bl6) and littermate *Trhr1*^−/− mice without (PVN^Con, *gray*) and with injection of AAV-TRH-hM3D-mCherry into the PVN (PVN^3D, *red*). TSH plasma concentrations are shown as percentage of Bl6 for normalization, because of a slight but non-significant increase of TSH in *Trhr1*^−/− compared to Bl6 mice. Two-way ANOVA for genotype: $F(1/20) = 5.3$, $p = 0.032$; *$p < 0.05$ (Bonferroni post test); mean ± S.E.M.; *n* as indicated. **f** Relative change in *Fos* mRNA expression ($2^{-\Delta\Delta CT}$) in the pituitary 45 min after CNO administration to Bl6 and *Trhr1*^−/− mice without (PVN^Con, *gray*) and with injection of AAV-TRH-hM3D-mCherry into the PVN (PVN^3D, *red*). Two-way ANOVA for genotype: $F(1/19) = 4.4$, $p = 0.049$; *$p < 0.05$ (Bonferroni post test); mean ± S.E.M.; *n* as indicated. **g** and **h** Two weeks after bilateral injection of AAV-TRH-hM3D-mCherry (PVN^3D) in the PVN of C57Bl/6 mice (PVN^3D-Bl6) hM3D-mCherry-positive fibers were detected in the ME. **g** Colocalization of hM3D-mCherry with ppTRH in fibers of the ME (*red*, hM3D-mCherry; *green*, ppTRH; *blue*, DAPI; scale bar, 50 μm). **h** Colocalization of hM3D-mCherry with CRH in fibers of the ME (*red*, hM3D-mCherry; *green*, CRH; *blue*, DAPI; scale bar, 50 μm). **i** Plasma TSH concentrations 2 h after CNO stimulation of PVN^Con-Bl6 (*gray*) and PVN^3D-Bl6 (*red*) mice. *$p = 0.0002$ (two-tailed Student's *t*-test); mean ± S.E.M.; *n* as indicated. **j** Plasma ACTH concentrations 2 h after CNO stimulation of PVN^Con-Bl6 (*gray*) and PVN^3D-Bl6 (*red*) mice. $p = 0.11$ (Student's *t*-test); mean ± S.E.M; *n* as indicated. **k–m** No colocalization of hM3D-mCherry with vasopressin (*green*, **j**), oxytocin (*green*, **k**), or GnRH (*green*, **l**) in fibers of the ME. *Red*, hM3D-mCherry; *blue*, DAPI; scale bar, 50 μm

This combination has the additional advantage that recombination is tamoxifen-inducible. In previous studies, *GlastCreER*^T2 mice were reported to selectively modify α-tanycytes without obvious recombination in β-tanycytes[40]. However, *Glast* is expressed in all subtypes of tanycytes[26] and we were able to recombine the Flex system and to express hM3D-mCherry also in β-tanycytes. Overall, the toolbox that we have developed will be helpful to analyze tanycyte functions in vivo. In the literature, a similar strategy has been reported that uses the protein tat-Cre[41] to label tanycytes[42] or induce tanycytic knockouts[32]. The advantage of our approach is

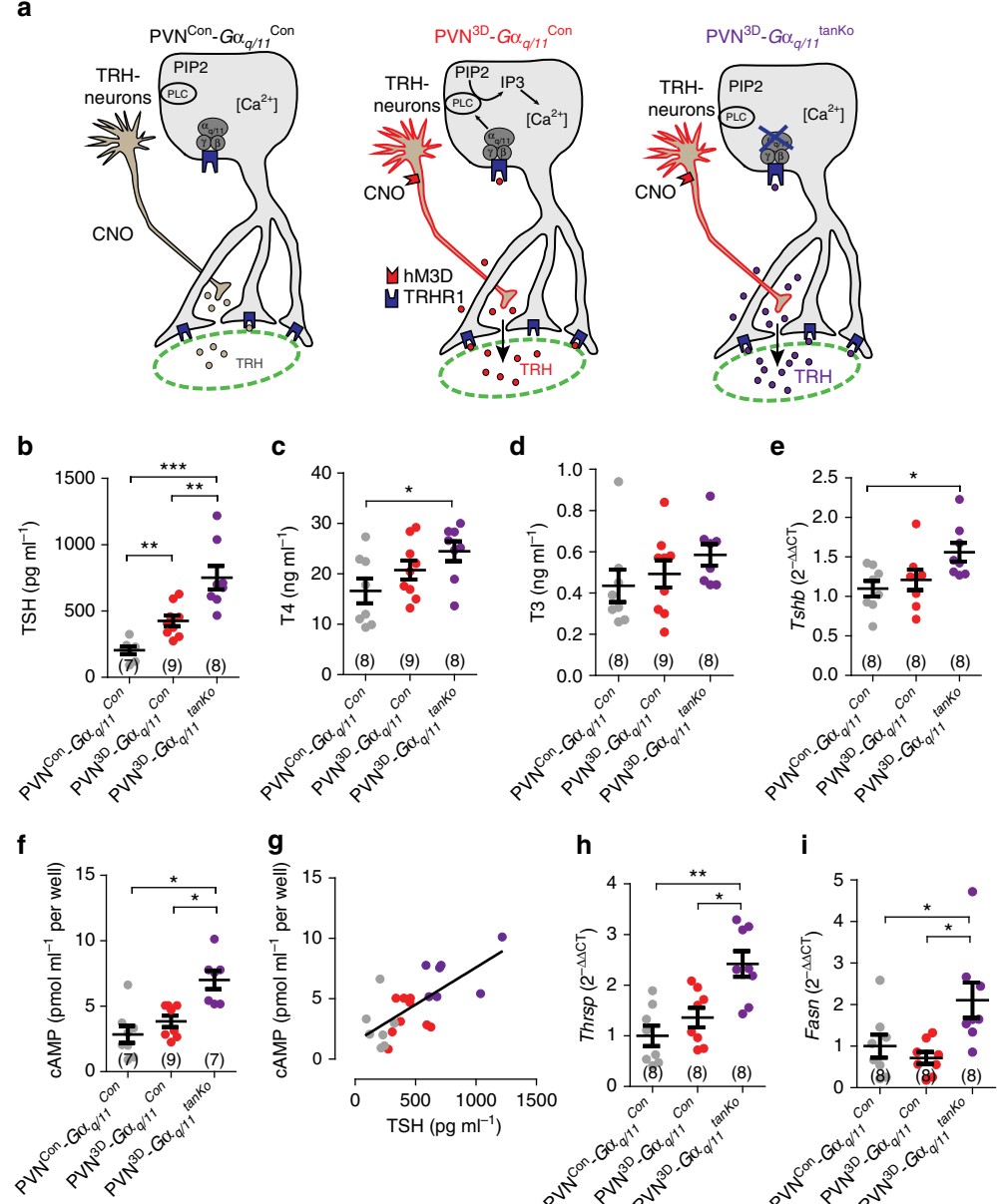

**Fig. 7** The tanycyte-specific deficiency of $G\alpha_{q/11}$ enhances the TSH release in response to activation of TRH neurons. **a** Schematic representation of the experimental groups. $PVN^{Con}-G\alpha_{q/11}^{Con}$: $G\alpha_q^{fl/fl}::G\alpha_{11}^{+/-}$ mice were injected with AAV-Dio2-GFP into the LV ($G\alpha_{q/11}^{Con}$) without injection of an AAV into the PVN ($PVN^{Con}$; gray). $PVN^{3D}-G\alpha_{q/11}^{Con}$: the PVN of $G\alpha_{q/11}^{Con}$ mice were bilaterally injected with AAV-TRH-hM3D-mCherry ($PVN^{3D}$, red). $PVN^{3D}-G\alpha_{q/11}^{tanKO}$: a deficiency of the $G\alpha_{q/11}$ pathway in tanycytes was induced by injecting AAV-Dio2-iCre-2A-GFP in the LV of $G\alpha_q^{fl/fl}::G\alpha_{11}^{-/-}$ mice. In parallel, AAV-TRH-hM3D-mCherry was bilaterally injected into the PVN to generate $PVN^{3D}-G\alpha_{q/11}^{tanKO}$ mice (purple). All mice received CNO. **b–e** TRH-dependent parameters 2 h after CNO (0.5 mg kg$^{-1}$ body weight) administration to $PVN^{Con}-G\alpha_{q/11}^{Con}$, $PVN^{3D}-G\alpha_{q/11}^{Con}$, and $PVN^{3D}-G\alpha_{q/11}^{tanKO}$ mice. **b** TSH plasma levels. One-way ANOVA, $F(2/22) = 23.6$, $p < 0.0001$. **$p < 0.01$, ***$p < 0.005$ between indicated groups (Bonferroni post test). **c** T4 plasma levels. One-way ANOVA, $F(2/24) = 3.4$, $p = 0.05$. *$p < 0.05$ between indicated groups (Bonferroni post test). **d** T3 plasma levels. One-way ANOVA, $F(2/23) = 3.7$, $p = 0.042$. $p > 0.05$ between groups (Bonferroni post test). **e** *Tshb* mRNA expression in the pituitary. One-way ANOVA, $F(2/23) = 4.3$, $p = 0.027$. *$p < 0.05$ between indicated groups (Bonferroni post-test). **f** As a measure of TSH activity, the intracellular cAMP concentration in CHO JP26 cells was determined 1h after stimulation with plasma of $PVN^{Con}-G\alpha_{q/11}^{Con}$, $PVN^{3D}-G\alpha_{q/11}^{Con}$, or $PVN^{3D}-G\alpha_{q/11}^{tanKO}$ mice. Plasma was collected 2 h after CNO stimulation. One-way ANOVA, $F(2/22) = 12.7$, $p = 0.0003$. *$p < 0.05$ between indicated groups (Bonferroni post test). **g** Linear correlation of TSH plasma concentrations (from **b**) and TSH activity expressed as stimulated cAMP concentrations (from **f**). $R^2 = 0.5861$, $p < 0.0001$ (Pearson correlation). **h** *Thrsp* mRNA expression in the liver. One-way ANOVA, $F(2/21) = 4.3$, $p < 0.0004$. *$p < 0.05$, **$p < 0.001$ between indicated groups (Bonferroni post test). **i** *Fasn* mRNA expression in the liver. Kruskal–Wallis test, 10.76, $p < 0.0046$. *$p < 0.05$ between indicated groups (Dunn's post test). mean ± S.E.M.; n as indicated

that we were able to overexpress genes, e.g., hM3D-mCherry, specifically in tanycytes.

With the tanycyte-specific rAAV-based Cre expression, we selectively deleted $G\alpha_{q/11}$ proteins in tanycytes to inhibit their TRH-induced activation. The tanycyte-specific deletion of $G\alpha_{q/11}$

proteins had no effects on basal plasma levels of TSH and thyroid hormones. However, the acute activation of the hypophysotropic TRH neurons had a stronger effect on plasma TSH levels when the $G\alpha_{q/11}$ pathway was inactivated in tanycytes. The fact that we did not see a basal elevation of thyroid hormones, when TRH

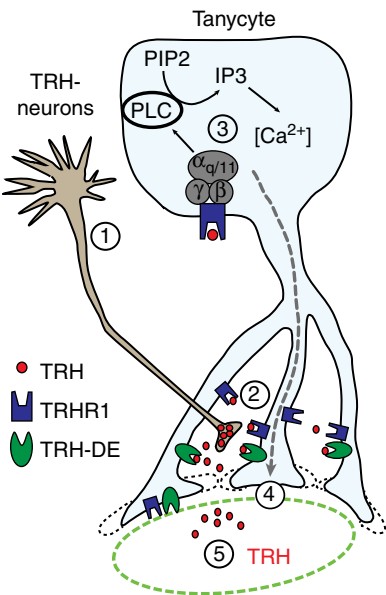

**Fig. 8** Schematic illustration of the tanycyte effects on TRH release in the ME. Activation of hypophysiotropic TRH neurons in the PVN (1) leads to TRH release in the ME (2). TRH stimulates the TRHR1-dependent $G\alpha_{q/11}$ pathway, which results in activation of the phospholipase C and increase in $[Ca^{2+}]_i$ (3). Activation of the tanycytic $G\alpha_{q/11}$ pathway enlarges tanycytic endfeet, which may retain TRH in the ME (4) and increases the time for degradation of TRH by the upregulated TRH-DE. Overall, activation of tanycytic $G\alpha_{q/11}$ signaling results in a downregulation of the TRH release into the pituitary vessels (5)

signaling was inhibited in tanycytes, may partially be explained by compensatory mechanisms, which seem to be overridden by acute activation of the HPT axis.

Two mechanisms may explain the increase in TSH release after acute stimulation of TRH neurons. First, tanycytes in the ME are in close proximity to TRH-positive nerve terminals and express TRH-DE[12, 13, 43]. This membrane-bound enzyme degrades TRH and seems to be an important modulator of the TRH release into the portal vessels of the pituitary[12, 43]. TRH-DE activity depends on $G\alpha_{q/11}$ signaling in tanycytes as the TRH-DE activity was reduced in $G\alpha_{q/11}^{tanKO}$ mice. The stimulation of TRH-DE activity by $G\alpha_{q/11}$ signaling in tanycytes is in contrast to in vitro experiments in pituitary cells, where TRH reduces the TRH-DE activity[44] suggesting cell-specific regulation. The reduced TRH-DE activity in $G\alpha_{q/11}^{tanKO}$ mice is a potential candidate to explain the increase of TSH levels after activation of TRH neurons.

A second possible mechanism relates to morphological changes in the ME. We observed a change in morphology of the tanycytic endfeet when we activated $G\alpha_{q/11}$ signaling in tanycytes. By expressing the membrane-bound hM3D-mCherry fusion protein in tanycytes, we were able to visualize the endfoot morphology. One hour after activating the $G\alpha_{q/11}$ pathway in tanycytes the tanycytic endfeet were larger, an effect that was abolished in $G\alpha_{q/11}^{tanKO}$ mice. Fast morphological changes are described in astrocytes in response to increased intracellular calcium[45]. In astrocytes vimentin and GFAP are phosphorylated via calcium-dependent kinases[46]. Both proteins are expressed in tanycytes[36]. Phosphorylation of these proteins leads to a change in the cytoskeleton and morphological changes in astrocytes[46–48]. Increasing of intracellular calcium waves also promotes movement of radial glia[49]. Thus, a change in tanycytic morphology by $G\alpha_{q/11}$ signaling and elevated $[Ca^{2+}]_i$ is not without precedent from glial cell biology. As for tanycytes, Parkash et al.[32] showed a morphological alteration in the ME via a semaphorin

7a-dependent mechanism. This leads to a tightening of the endfoot structure of the tanycytes covering the fenestrated vessels of the ME as well as a retraction in GnRH neuronal terminals. Thereby, tanycytes in the ME seem to regulate GnRH secretion[32].

Our data show that tanycytes limit the TRH release from the hypothalamus and thus control the response of the HPT axis to an acute activation of TRH neurons. Interfering with their function led to higher T4 and T3 plasma concentrations but also induced the expression of T3-dependent genes in the liver. This tanycytic mechanism may play a homeostatic function during chronic hypo- or hyperactivity of the HPT axis. Loosening the tanycytic brake during hypothyroidism or tightening it during hyperthyroidism will stabilize the axis. In addition tanycytes may play a role in shaping the pulsatile pattern of TSH secretion[50]. Until now it is unknown how exactly the pulsatile release of TSH is regulated, although there is evidence that the pulsatile TSH secretion pattern is mediated by TRH[51]. In rats, TRH is released into the portal vein system in a pulsatile manner[52]. A combination of morphological changes of tanycytes and regulation of the TRH-DE activity could be responsible for pulsatile release.

In summary we have identified a new mechanism by which tanycytes control the HPT axis (Fig. 8). After release from neuronal terminals in the ME, TRH activates TRHR1 and the coupled $G\alpha_{q/11}$ pathway in tanycytes. This pathway induces structural changes in tanycytic endfeet, which increase their size. In addition, the $G\alpha_{q/11}$ pathway in tanycytes modulates the TRH-DE activity. We suggest that by changing their morphology, tanycytes impede the vascular outflow of TRH and instead favor its degradation via TRH-DE. Thereby, tanycytes are able to modulate the HPT axis at the hypothalamic level.

## Methods

**Mice.** All mouse lines were established on a C57Bl/6 background. We used male littermate mice that were age-matched between experimental groups. Mice were between 8 and 14 weeks of age. All animal experiments were approved by the local animal ethics committee (Ministerium für Landwirtschaft, Umwelt und ländliche Räume, Kiel, Germany). Mice were kept at constant temperature (22 °C) on a 12-h light/dark cycle and were provided with standard laboratory chow (2.98 kcal g⁻¹; Altromin, Hannover, Germany) and water ad libitum.

Mice that carried a floxed $G\alpha_q$ allele ($G\alpha_q^{fl/fl}$) and were $G\alpha_{11}$-deficient ($G\alpha_{11}^{-/-}$)[23] ($G\alpha_q^{fl/fl}::G\alpha_{11}^{-/-}$) were used to knockout the $G\alpha_{q/11}$ pathway in tanycytes by transducing tanycytes with the cell-specific AAV-Dio2-iCre-2A-GFP vector of serotype 1/2 ($G\alpha_{q/11}^{tanKO}$). $G\alpha_q^{+/-}$ ($G\alpha_q^{fl/fl}::G\alpha_{11}^{+/-}$) littermates that received an injection of AAV-Dio2-GFP of serotype 1/2 ($G\alpha_{q/11}^{Con}$) were used as controls. To delete $G\alpha_{q/11}$ signaling in glial cells, $GlastCreER^{T2}$ mice[25] were crossed with animals that carried a floxed $G\alpha_q$ allele ($G\alpha_q^{fl/fl}$) on a $G\alpha_{11}$-deficient ($G\alpha_{11}^{-/-}$) background[23] ($G\alpha_{q/11}^{gliaKO}$). $GlastCreER^{T2}$ mice[25] were also used to express the hM3D-mCherry specifically in tanycytes. For this purpose, we injected the Cre-dependent AAV-CAG-flex(hM3D-mCherry) vector into the lateral ventricle of mice heterozygous for the Cre allele (gTan³D). To induce CreER^{T2} activity in transgenic animals at an age of 8–10 weeks, they were treated i.p. with 1 mg tamoxifen dissolved in 90% miglyol 812, 10% ethanol every 12 h for 5 consecutive days. Generation and phenotyping of Trhr1 knockout mice (Trhr1⁻/⁻) have been reported previously[53, 54]. Trhr2 knockout mice (Trhr2⁻/⁻) were obtained from Deltagen. For chemogenetic stimulation of hypophysiotropic neurons in the PVN, AAV-TRH-hM3D-mCherry was injected bilaterally into the PVN (PVN³D). To test the Cre activity of rAAV vectors, we used the reporter mouse line Ai14[55]. All mice were randomly allocated to treatment groups. Investigators were blinded for treatment or genotype of mice or both in all experiments and analyses. Mice were only excluded from analysis if they did not survive during surgical procedures or if no blood samples could be obtained.

**Cloning of viral vectors.** rAAV gene transfer vectors were generated by inserting specific gene cassettes into rAAV2-based expression vectors, containing inverted terminal repeats of serotype 2[56], a woodchuck posttranscriptional regulatory element (WPRE), and a bovine growth hormone polyadenylation site (bGHpA). The plasmid pAAV-Dio2-iCre-2A-GFP was generated by PCR amplifying the small upstream fragment of the human DIO2 gene (−1757 to +157, GenID: 1734) from human complimentary DNA (cDNA) and inserting it in the Xho1/Age1 site of pAAV-CMV-iCre-2A-GFP[57] to replace the CMV promoter. The plasmid pAAV-TRH-hM3D-mCherry was generated by PCR amplifying of the hM3D-mCherry sequence from the plasmid pAAV-Syn-flex(hM3D-mCherry)[24] and inserting it in frame in pAAV-CAG-Venus[58] in order to replace Venus. In a second

step, the CAG promoter was exchanged for a short TRH promoter fragment (−776 to +84; GenID: 25569)[29] amplified from rat cDNA. The plasmid pAAV-CAG-flex(hM3D-mCherry) was generated by inserting the CMV enhancer chicken β-actin (CAG) promoter from pAAV-CAG-BMP2-2A-Tomato[58] into the Mlu1/Sal1 site of pAAV-Syn-flex(hM3D-mCherry)[24] to replace the promoter. The plasmid pAAV-CAG-GCamP6s[18] was generated by inserting the CGaMP6s sequence of pGP-CMV-GCaMP6s (Plasmid #40753, Addgene) into the Nhe1/Not1 site of pAAV-CAG-Venus[58] to replace the Venus sequence. After construction, all plasmids were sequenced.

**Recombinant AAV production**. rAAV with a mosaic capsid of serotype 1 and 2 (1:1) were generated as described[57] and purified by AVB Sepharose affinity chromatography[59]. For each vector, the genomic titer was determined by quantitative PCR (qPCR) using primers against WPRE (WPRE forward primer: 5′-TGCCCGCTGCTGGAC-3′; WPRE reverse primer: 5′-CCGACAACACCACG GAATTG-3′) as described previously[58].

**Stereotaxic injections**. Stereotaxic rAAV injections were performed according to a protocol described previously[60]. The following coordinates relative to bregma were used for injections: lateral ventricle, anteroposterior −0.1 mm, mediolateral −0.9 mm, dorsoventral from the skull surface −2.3 mm; PVN, anteroposterior −0.7 mm, mediolateral ±0.25 mm, dorsoventral from the skull surface −4.9 mm. Mice were anesthetized with an intraperitoneal (i.p.) injection of a mixture of ketamine hydrochloride (65 μg g⁻¹ body weight) and xylazine (15 μg g⁻¹ body weight) in NaCl (0.9%). After loss of reflexes, animals were fixed in a stereotaxic frame (David Kopf instruments; Nr: 1900). At the defined positions, small holes were made into the skull using a dental drill (freedom; K.1070 Micromotor Kit). Vector solution was injected into the PVN (120 nl per site) and into the lateral ventricle (2–2.5 μl) at a rate of 100 nl min⁻¹. After injection, the micropipette was kept in place for 5 min to avoid backflow of the injected vector during micropipette retraction. The scalp was sutured, and the animal was placed on a heating pad until full recovery from surgery and then returned to its home cage. Carprofen (5 mg kg⁻¹ body weight; s.c.) was applied once a day for 2 days after rAAV injection.

**Calcium measurement**. To detect changes of [Ca²⁺]ᵢ, tanycytes were transduced with the calcium sensor GCamP6s[18] by injecting AAV-CAG-GCamP6s (1.5 × 10¹⁰ genomic particles) into the lateral ventricle of mice 2 weeks prior to the measurements. Brains were dissected from deeply anesthetized mice (ketamine hydrochloride, 65 μg g⁻¹ body weight; xylazine, 15 μg g⁻¹ body weight) and placed in ice-cold artificial cerebrospinal fluid (aCSF; 124 mM NaCl; 26 mM NaHCO₃; 1.25 mM NaH₂PO₄; 3 mM KCl; 2 mM CaCl₂; 1 mM MgSO₄; 10 mM D-glucose). Brain slices (150–200 μm in thickness) were prepared under controlled conditions (aCSF + 10 mM MgSO₄, 5% CO₂, 95% O₂; pH 7.4; 4 °C) using a vibrating blade microtome (Leica VT1200 S). Slices were placed in normal aCSF at 4 °C for 45 min before being transferred to an ice-cold aCSF with low glucose (1 mM glucose + 9 mM sucrose). Slices could be stored up to 5 h at 4 °C without a reduction in reactivity.

To measure [Ca²⁺]ᵢ in response to CNO stimulation in gTan³ᴰ or gTan^Con mice, acute brain slices were loaded with Fura-2AM. Slices were incubated with Fura-2AM (12.5 μg ml⁻¹ with 0.5% DMSO and 0.05% pluronic 127, Invitrogen) in low-glucose aCSF for 30 min at 37 °C. Then, they were incubated in aCSF containing probenecid (1.25 mM, 30 min 37 °C) and stored at 4 °C before measurement.

[Ca²⁺]ᵢ was measured using a high-speed calcium imaging setup (Till Photonics) mounted on the Axio Examiner D1 upright fluorescent microscope (Zeiss) coupled to the polychrome V monochromator and a high-speed CCD camera (Retiga EXi-blue, QImaging). Data acquisition and quantification was done using life acquisition and offline analysis software (FEI GmbH Munich, formerly Till Photonics). Brain slices were placed in a flow chamber. For measuring GCamP6s, an excitation wavelength of 488 nm and an emission filter (> 498 nm) were used. Changes in fluorescence (F) were normalized to basal mean values of the first 30 s (F₀) of each region of interest (ROI) and were shown as (F/F₀). ROIs covered the whole tanycytic layer of the ME.

For single-cell calcium imaging, we used a confocal microscope (TCS SP5, Leica) with a 488 nm laser using a ×20 objective (HCX PL APO CS 20X). Changes in fluorescence (F) were normalized to basal mean values of the first 30 s (F₀) of each ROI and were shown as (F/F₀). ROIs covered the cell bodies of single tanycytes of the ME or tanycytes of the ependymal layer of the 3rd ventricle. F and F₀ were analyzed with Fiji.

For the ratiometric measurement of Fura-2, we used excitation wavelengths 340 and 380 nm. Changes in the fluorescence ratio of F₃₄₀/F₃₈₀ (ΔF) were normalized to basal mean values of the first 30 s (F₀) of each ROI and were shown as (ΔF/F₀). ROIs covered single tancytic cell bodies. hM3D-mCherry-positive tanycytes were identified by the red fluorescence of mCherry.

After 1 h of recovery in the measurement buffer (aCSF; CO₂, 5%; O₂ 95%; pH, 7.4; flow rate, 2 ml min⁻¹), stimuli were administered into the perfusion flow via a dedicated perfusion system (TSE-systems). One minute after starting the measurement, TRH (33 μM) or taltirelin (33 μM, 150 μM for single-cell response)

were administered for 15 s. ATP (33 μM, 150 μM for single-cell response, 15 s) was used as positive control.

For blocking the Gα_{q/11} signaling pathway, slices were stimulated once with taltirelin (33 μM, 15 s) followed by incubation with 2-APB (100 μM, Tocris) or U73122 (100 μM, Tocris) in aCSF for 30 min and afterwards a second taltirelin stimulation. As 2-APB and U73122 were dissolved in 0.5% DMSO, we treated control slices with aCSF containing 0.5% DMSO. For TRH receptor inhibition, after a first taltirelin stimulation midazolam (500 μM) was added for 20 min directly to the aCSF flow with a perfusion pump (TSE) followed by a second taltirelin stimulation (33 μM, 15 s) in addition to midazolam. After 20 min washout with aCSF (2 ml min⁻¹), a third taltirelin stimulation was performed. For quantification the maximal F/F₀ was used. For heat maps we used the Look-Up Table royal from Fiji, representing a gray sale from 0 to 255.

**Chemogenetic stimulation of TRH neurons and hormone measurements**. AAV-TRH-hM3D-mCherry (1.2 × 10⁸ genomic particles) was bilaterally injected into the PVN of mice (PVN³ᴰ). Mice without injection of the PVN (PVN^Con) were used as control. For the parallel knockout of the Gα_{q/11} pathway in tanycytes, AAV-Dio2-iCre-2A-GFP (2 × 10¹⁰ genomic partials, 2 μl) was injected into the lateral ventricle of Gα_q^{fl/fl}::Gα₁₁⁻/⁻ mice (PVN³ᴰ-Gα_{q/11}^{tanKO}) during the same surgery. Gα_q^{fl/fl}::Gα₁₁⁺/⁻ mice that received an injection of AAV-Dio2-GFP virus in the lateral ventricle (PVN³ᴰ-Gα_{q/11}^Con and PVN^Con-Gα_{q/11}^Con) were used as controls. Three weeks after rAAV injection, all three animal groups received CNO (0.5 mg kg⁻¹ body weight; i.p.). After CNO stimulation, plasma was collected for determining TSH, T4, and T3 concentrations using a MILLIPLEX mouse kit (Millipore Corp.) and a BioPlex-system (Bio-Rad) according to the manufacturers' instructions, while pituitary mRNA of Tshb and Fos was quantified by qPCR.

PVN³ᴰ-Trhr1⁻/⁻ and PVN³ᴰ-Bl6 as well as control animals (PVN^Con) were treated with CNO (0.5 mg kg⁻¹ body weight, i.p.) 3 weeks after rAAV injection. Plasma TSH was measured 45 min or 2 h after CNO stimulation. Pituitary mRNA of Tshb and Fos was quantified by qPCR.

**TRH-DE activity assay**. Gα_{q/11}^{tanKO} and Gα_{q/11}^Con mice were killed by decapitation and the medial basal hypothalamus (right and left arcuate nuclei plus ME) was microdissected with a stereomicroscope using small spring scissors (FST). Tissue was homogenized in a microcentrifuge tube with an ultra-turrax in 50 mM sodium phosphate buffer (pH 7.5) followed by centrifugation for 15 min at 14,000×g. Pellets were rinsed with 1 M NaCl and resuspended in 50 mM sodium phosphate buffer (pH 7.5). Membrane TRH-DE activity was determined using the substrate TRH-β-naphthylamide (TRH-βNA, Bachem) in a coupled enzyme assay in the presence of excess dipeptidyl aminopeptidase IV (ACRO biosystems; activity: 4 nmol Gly-Pro-βNA (Bachem) hydrolyzed per min) containing 0.2 mM N-ethylmaleimide (Sigma Aldrich), 0.2 mM bacitracin (Sigma Aldrich). The enzymatic reaction was initiated by addition of 800 μM TRH-βNA to the reaction mixture (total volume 300 μl) and carried out in duplicates at 37 °C. About 50 μl was withdrawn every 30 min over 120 min. The reaction was stopped by the addition of methanol (50 μl, 100%). Aliquots were diluted 1:20 with 50% sodium phosphate buffer (50 mM, pH 7.5, containing 50% methanol). β-NA was measured in a fluorometer (excitation 335 nm; emission 410 nm). An aliquot was kept for protein quantification using the Lowry method. The β-NA concentration was calculated by a standard curve diluted in 50% sodium phosphate buffer (50 mM, pH 7.5, containing 50% methanol). The activity was linear within the first 2 h and was analyzed relative to the membrane protein content for each individual sample.

**Bioassay of TSH activity**. TSH activity in plasma samples was measured with a bioassay, using CHO cells stably transfected with cDNA for a human TSH receptor (JP26-CHO, a gift of Gilbert Vassart)[33]. Cells (1 × 10⁴ per well in a 96-well plate) were starved for 30 min with Krebs Ringer HEPES buffer (KBH: 124 mM NaCl; 1.25 mM KH₂PO₄; 1.45 mM CaCl₂; 1.25 mM MgSO₄; 5 mM KCl; 25 mM Hepes; 8 mM D-glucose; 0.05% bovine serum albumin; pH 7.4) before they were stimulated with plasma samples (100 μl diluted 1:2 in KBH) for 1 h. At the end of the incubation time, cells were lysed by adding 100 μl 0.1 N HCl containing 0.5% Tween 20 and cAMP was measured with cAMP ELISA (Enzo) as described[61].

**Immunohistochemistry and analysis of endfoot morphology**. To visualize tanycytic endfeet, AAV-CAG-flex(hM3D-mCherry) was injected into the lateral ventricle of GlastCreER^T2 mice (Tan³ᴰ; 1 × 10⁹ genomic particles/mouse). One week after the injection, mice were treated with tamoxifen as described above. Two weeks after vector transduction, mice were stimulated with CNO (0.5 mg kg⁻¹ body weight; i.p.) and after 60 min perfused with 4% paraformaldehyde (PFA) during deep anesthesia (ketamine hydrochloride, 65 μg g⁻¹ body weight; xylazine, 15 μg g⁻¹ body weight). Brains were postfixed in 4% PFA at 4 °C overnight. Slices (50 μm) were prepared with a vibratome. Free-floating sections were washed twice in Tris-buffered saline (TBS), permeabilized with 0.3% Triton X-100 in TBS (TBS-TX) for 30 min, and incubated in bovine serum albumin (5% in TBS-TX) for 2 h. Then, primary antibodies were added followed by overnight incubation with gently shaking. Sections were washed twice in TBS-TX for 10 min and incubated for 2 h with secondary antibodies. After washing twice in TBS, sections were mounted on glass slides and covered with Mowiol. Endfoot sizes were analyzed using a confocal

microscope (SP5, Leica; objective, HCX PL APO CS 63X oil UV corrected; aperture, 1.4; microscopic zoom, 5 to ×13; scanning frequency, 100 Hz; average: 4 times and pinhole, 0.5 AU with a z-stack over 20 µm with a step size of 0.5 µm). Pictures were analyzed by Fiji. Per animal 15–32 endfeet in five areas of three slices were analyzed. For costainings with anti-vimentin mice were transcardially perfused with 4% PFA. Subsequently, the brains were snap-frozen in 2-methylbutane on dry ice and were sectioned with a cryostat. Staining was performed using the protocol described above. Primary antibodies: mCherry (1:1,000, Acris #AP32117-PU-N), vimentin (1:250, Thermo Fisher #PA1-16759), 2A (1:1000, Millipore #ABS31), collagen IV (1:1000, Abcam #ab6586), ppTRH160–169 (1:2000, gift from M. Wessendorf), GnRH (1:500, immunoStar #20075), vasopressin (1:500, immunoStar #20069), corticotropin-releasing hormone (1:500, immunoStar #20084, CRH), oxytocin (1:500, immunoStar #20064), aquaporin 4 (1:1000, Millipore #AB3594), cFos (1:500, Abcam #ab7963), and neurofilament 200 (1:500, Sigma Aldrich #N-4142). Secondary antibodies: anti-rabbit alexa 488 (1:500, Invitrogen #A-21206) anti-chicken Cy5 (1:500, Abcam #ab97147), anti-goat Cy3 (1:500, Jackson ImmunoResearch #705-165-147). DAPI was used in a concentration of 1 µg ml$^{-1}$.

In other experiments, we injected AAV-CAG-flex(hM3D-mCherry) and AAV-Dio2-iCre-2A-GFP ($2 \times 10^{10}$ genomic particles of each vector, 2.5 µl) into the lateral ventricle of $G\alpha_q^{fl/fl}$::$G\alpha_{11}^{-/-}$ mice to generate dTan$^{3D}$-$G\alpha_{q/11}^{tanKO}$ and of C57Bl/6 mice to generate dTan$^{3D}$-Bl6. dTan$^{3D}$-Bl6 mice were used as controls. Two weeks later, mice were treated with taltirelin (1 mg kg$^{-1}$ body weight; i.p.), CNO (0.5 mg kg$^{-1}$ body weight; i.p.), or saline (5 µl g$^{-1}$ body weight; i.p) and brains were dissected and analyzed as described above.

**In situ hybridization.** To determine the localization of *Trh* mRNA expressing neurons and hM3D-mCherry in PVN$^{3D}$ mice, free-floating in situ hybridization was performed. Animals were killed in deep anesthesia (ketamine hydrochloride, 65 µg g$^{-1}$ body weight; xylazine, 15 µg g$^{-1}$ body weight) and brains were postfixed in 4% PFA at 4 °C overnight. Sections (50 µm) were prepared with a vibratome and hybridized with a digoxigenin-labeled anti-*Trh* RNA probe[62] (530 bp, bases 202–733 of NM_009426) overnight. Hybridization was detected with anti-digoxigenin-POD Fab-fragment antibodies (Roche) at a dilution of 1:100 overnight. The POD signal was increased with the Tyramin Cy5 amplification kit (PerkinElmer, 1:100). After in situ hybridization, hM3D-mCherry was detected by immunohistochemistry with anti-mCherry antibodies (1:500, Acris #AP32117-PU-N).

**Quantitative PCR.** RNA was isolated from pituitary glands and liver by using the Qiagen Nucleospin 96 kit according to the manufacturer's instructions. First strand cDNA was synthesized using oligo-(dT)15 primer and AMV Reverse Transcriptase (RT; Invitrogen). cDNA was stored at −20 °C until further analysis. The following primers were used for quantitative real-time PCR (qPCR): *Tshb* forward, 5′-CCGCACCATGTTACTCCTTA-3′, *Tshb* reverse, 5′-GTTCTGACAGC CTCGTGTAT-3′, PCR product 104 bp; *Fos* forward, 5′-CTTTCCCCAAACTTC-GACCA-3′, *Fos* reverse, 5′-TCGTAGTCGGCGTTGAAACC-3′, PCR product 51 bp; *Gapdh* forward, 5′-ATGTGTCCGTCGTGGATCTGA-3′, *Gapdh* reverse, 5′-TGAAGTCGCAGGAGACAACCT-3′, PCR product 144 bp; *Thrsp* forward, 5′-CTTACCCACCTGACCCAGAA-3′, *Thrsp* reverse 5′-CATCGTCTTCCCTCTC GTGT-3′ PCR product 120 bp; *Fasn* forward, 5′-GGAGGTGGTGATAGCC GGTAT-3′, *Fasn* reverse 5′-TGGGTAATCCATAGAGCCCAG-3′, PCR product 140 bp. qPCR was performed according to the following protocol: 2 min at 50 °C, 2 min at 95 °C, 15 s at 95 °C, and 1 min at 60 °C (40 cycles). Amplification was quantified using Platinum SYBR Green qPCR SuperMix (Invitrogen). Quantified results were normalized to *Gapdh* using the ΔΔCt method.

**Trhr1 PCR.** Cryosections of C57Bl/6 mice (20 µm in thickness) were mounted on glass slides covered with a membrane of polyethylene naphthalate (1.35 µm in thickness, Zeiss, Germany) and stored at −80 °C. Alpha- and β-tanycytes were microdissected using a system with a pulsed 337-nm UV laser (PALM MicroBeam; PALM Microlaser Technologies) and the laser-pressure catapulting mode. RNA was isolated with Absolutely RNA Nanoprep Kit (Agilent). cDNA was amplified as described for qPCR. A two-step PCR was performed (20 cycles followed by 30 cycles) with the following primers: *Trhr1* forward, 5′-CAGCACCTACAAAAAC GCTG-3′; R-*Trhr1* reverse, 5′-CTTTATGGCCTTCTTTAGTTCTCAG-3′; fragment size: 693 bp; *Actb* forward, 5′-ATGGAATCCTGTGGCATCCAT-3′; *Actb* reverse, 5′-TTCTGCATCCTGTCAGCAATG-3′; fragment size: 140 bp.

**Statistical analysis.** All analyses were performed using Prism 5 (GraphPad Software). Data were assessed for normality (Shapiro–Wilk test) and equality of variances (Bartlett's test or Levene test). If the raw data did not satisfy these conditions, a non-parametric method was used. Data of two groups were compared by a two-tailed, unpaired, parametric Student's *t*-test. For parametric multiple comparisons, we used one- or two-way ANOVA with Bonferroni post hoc test. Kruskal–Wallis with post hoc Dunn's test was used for non-parametric multiple comparisons analysis. The significance level was set at $p < 0.05$. All data are shown as means ± S.E.M. G-Power (Version 3.1.8) was used to analyze power and group size before the experiments.

**Data availability.** All the relevant data are available from the authors upon request.

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

## Acknowledgements

We thank Ines Stölting, Andreas Makurat for technical assistance, Magdalena Goetz for providing the *GlastCreER*^T2 mouse line, Martin Wessendorf for providing us the ppTRH-Antibody, Gilbert Vassart for the JP26-CHO cell line. We also thank Rolf Sprengel for providing the plasmid pAAV-CMV-iCre-2A-GFP and pAAV-CAG-BMP2-2A-Tomato as well as Bryan Roth for the pAAV-Syn-hM3D-mCherry plasmid. This work was supported by grants from the Deutsche Forschungsgemeinschaft (SPP1629 "Thyroid Trans Act", MU 3743/1-1 to HMF; GRK1957 "Adipocyte-Brain Crosstalk", and TRR134 to MSch).

## Author contributions

H.M.-F: Involved in most of the experiments and analyzed the results; M.S., S.A., and V.K.: Performed calcium measurements; H.M.-F., A.B., and V.K., Plasmid generation; M.S., H.M.-F., and A.B., Immunohistochemical stainings and image analysis; M.R. and J.M.: In situ hybridization; H.M.-F. and J.W.: Animal experiments; M.B. and K.K.: Microdissection; M.S., S.A., and M.R: PCR; H.H. and S.O.: Provided valuable reagents and contributed to the manuscript. H.M.-F. and M.Sch.: Designed and supervised the study and wrote the manuscript.

## Additional information

**Competing interests:** The authors declare no competing financial interests.

