## [Peer Review File · Nature Communications]

Reviewers' expertise:

Reviewer #1: Hypothalamus tanycyte, metabolism;

Reviewer #2: Neural circuits in the hypothalamus, systems neuroscience approaches;

Reviewer #3: Nuclear hormone receptors, signal transduction.

Reviewers' comments:

Reviewer #1 (Remarks to the Author):

The manuscript by Muller-Fielitz and colleagues explores novel regulatory processes controlling neurosecretion operating at the tanycytic endfeet of the gliovascular unit in the median eminence, a neurohemal organ located in the ventral part of the tuberal region of the hypothalamus. In particular, using cutting-edge approaches and original virogenic tools, the authors provide the first in vivo demonstration that tanycytes control the bioavailability of the thyroid-releasing hormone (TRH) and its release into the pituitary portal blood vessels, when TRH neurons are activated. This is a conceptually important and logical manuscript. Technically, it is a "tour de force" compilation of many advanced techniques, which clearly increase its appeal. These methods are sound and sufficient to support the overarching hypothesis. The paper certainly is an important piece of work, which would feature well in Nature Communications.

The manuscript reads beautifully and has been carefully prepared. My specific comments are elaborated below:

1. The authors show that by chemogenetically activating G-alpha-q/11 in tanycytes using the DREADD technology in a GlastCreERT2 mouse background, they change the morphology of tanycyte endfeet within one hour after CNO administration. The efficiency of the tamoxifen-induced cre-mediated gene recombination being highly variable from one animal to another, how did the authors control the fact that the changes of the expansion of tanycytic endfeet they see in Figure 2d-f was actually due to the effect of CNO? It would be more convincing if the authors could complement these data by experiments using living brain slices in which mCherry-positive tanycytic endfeet could be imaged before and after CNO application into the bath (i.e., like they did in Figure 1 and 2C). An alternative, would be to show in Figure 2d and 2e a lower magnification image illustrating the fact that the same number of tanycytes have been transduced in each condition (i.e., number of mCherry-positive tanycytic cell bodies lining the wall of the third ventricle).
2. The authors convincingly demonstrate that manipulating tanycyte function tightly modulates the activity of the hypothalamic-pituitary-thyroid hormone axis when TRH neurons are active, as measured by TSH release and activity. Are there any physiological and pathological states in which manipulating tanycytic function could alter the adaptive response of the thyroid axis and result in measurable changes in metabolism (e.g., changes in temperature)? This point can be touched upon in the discussion.

Minors:

i) Line 317, what is the serotype of the AVV used?

ii) Line 324, at what age were the animals injected with tamoxifen? If different ages have been used, was there any changes in treatment efficiency between ages?

Reviewer #2 (Remarks to the Author):

Tanycytes are found in close apposition to the endings of TRH neurons in the median eminence raising the possibility that they may play an important role in controlling the output of the hypothalamic-pituitary-thyroid axis (HPT). The role of tanycytes in regulating the release of GnRH has been investigated, but to the best of my knowledge, this is the first exploration for a role in TRH release.

The authors use a number of different approaches to show that activation of TRH1 receptors on tanycytes modulates HPT output. They propose a model whereby tanycytes inhibit the HPT. This requires activation of TRH1 receptors on tanycytes which increases Ca, causing an increase in the size of the endfeet

Each of the observations taken independently, is interesting. I do, however, have concerns about the use of the tools. In particular, I'm puzzled by the multiple mouse lines and am not convinced about the specificity of the targeting. In addition, since the mechanistic experiments are conducted in vitro, I would like to see a higher level of sophistication and resolution for a journal like Nature Communications. In particular, I think single-cell resolution for Ca changes should be required. I also think the data, as presented, for the endfoot volume changes makes it difficult for readers to discern the changes for themselves.

Major Concerns

- 1) I find the use of multiple mouse lines confusing. For example, the initial examination of TRHR1-dependent increases in endfoot size is done in GlastCreERT2 mice. Then, the authors show that the change in endfoot size induced by taltirenin is eliminated by genetic elimination of Gq11a in Tan3D mice. Why the different strains? Unless I have missed something, I do not see the need for using the GlastCreERT2 mice.
- 2) Additionally, I am concerned about potential contributions from astrocytes in the GlastERT2. The authors are aware of this as they show images in supplementary data that are consistent with little/no expression in astrocyte. While this makes a stronger case for tanycytes, I'm puzzled that there is no overlap with astrocytes.
- 3) The changes in Ca in tanycytes shown in figure 1 is a good starting point. A cellular-level resolution of the Ca signal would be useful here. Also, it's not clear how the data were quantified. Is it better to examine the peak or integrate the Ca signal? Single-cell resolution, 2-photon imaging would add substantially to these observations.
- 4) The CNO experiment (n=4) is underpowered. I would feel more confident with additional experiments to solidify this observation.
- 5) The supplementary figures look beautiful, but there is no quantification. It is imperative that there is selective targeting of tanycytes with the construct. Whilst this appears to be the case, the reader cannot be certain without quantification.

Minor

Why were plasma TSH concentrations measured only 2 hours after stimulation of TRH cells? A better temporal quantification of changes in TSH would be helpful.

I'm concerned by the mcherry expression in TRH neurons in Fig 5C.

The abstract states that tanycytes are located in close proximity to TRH neurons. This is not accurate. The neurons (cell bodies) are found in the PVN and the axon terminals of these neurons are in the median eminence, adjacent to tanycytes.

Reviewer #3 (Remarks to the Author):

The hypothalamic-pituitary-thyroid (HPT) axis is normally tightly regulated so as to maintain thyroid hormone levels at the correct physiological concentrations. This manuscript reports a new potential role for hypothalamic β 2-tanycytes in influencing this homeostatic regulation of the HPT axis. The authors report that thyrotropin-releasing hormone (TRH), operating through a TRH receptor-1/Gaq/11 mediated mechanism in the β -tanycytes in the median eminence, produces increases in the size of the tanycyte endfeet that project onto the pituitary vessels and increases in the activity of the TRH-degrading ectoenzyme TRH-DE. These events would be expected to impair thyroid stimulating hormone (TSH) release by the TRH-responsive pituitary cells in the median eminence through physical shielding and/or by increased TRH degradation. Consistent with this prediction, the authors report that blocking TRH signaling in tanycytes by ablating Gaq/11 expression in these cells enhances the response of the HPT axis to the chemogenetic activation of TRH neurons. The authors conclude that their study identifies "new TRH- and Gaq11-dependent mechanisms in the median eminence by which tanycytes control the activity of the HPT axis."

Overall this is a fairly interesting study. The identification of a possible role of tanycytes in modulating TRH availability is itself intriguing and would not only reveal an unanticipated regulation of the HPT axis but would also support the more general concept that tanycytes can function as modulators of a variety of endocrine signaling pathways. The authors convincingly establish that TRH/taltirelin can alter Gaq/11-calcium signaling in β 2-tanycytes, that chemogenetic stimulation of Gaq/11 pathways in these tanycyte cells (by expression of ectopic hM3D and CNO treatment) results in moderate increases in the diameters of the endfeet of these cells in a Gaq/11 dependent fashion, and that chemogenetic stimulation of TRH release by the ME-associated TRH neurons produces enhanced serum levels of TSH which are further increased when Gaq/11 signaling was ablated in the tanycytes. Whereas taltirelin (a TRHR agonist) increased TRH-DE enzymatic levels in control mice, ablating Gaq/11 signaling in the tanycytes resulted in decreased levels of TRH-DE enzymatic activity and in smaller endfeet (the latter when compared to control mice with both assayed in the presence of taltirelin).

These individual observations are valuable and are consistent with the authors' overall conclusion that β -tanycytes, in response to TRH, can alter/regulate the availability of this hormone in the ME. However some question still remains if these individual experimental observations add together sufficiently to completely support the authors' overall conclusion, and if they do, how substantial a contribution these phenomena actually represent in the physiological regulation of the HPT axis. Further, much of the analysis in this manuscript relies heavily on the use of fairly complex combinations of genetically manipulated mice, the introduction of a variety of ectopic CRE and reporter constructs into anatomically specific regions of the brain, and the use of non-physiological chemical inducers and inhibitors of specific pathways. The technical complexity of these methodologies raises some issues as to the biological relevance of the results obtained, and although the manuscript is well written, also makes it somewhat difficult for an average reader to easily follow all the logic and implications of the experiments. The authors should address these issues in more detail. Specific concerns and suggestions are detailed below:

1. As noted above, some of the evidence provided that tanycytes regulate the HPT axis is indirect. For example, although acute stimulation through the CNO/hM3D approach detectably altered serum TSH levels and this was further enhanced by ablating Gaq/11 signaling in the tanycytes (Figure 6), acute treatment with TRH itself (or a similar agonist) increased TRH-DE levels without a reported effect on TSH plasma levels; also, abolishing Gaq/11 signaling in these cells resulted in no detectable change in steady state serum T3, T4, or TSH levels despite reducing the associated TRH-DE levels (Figure 3, the

lack of which the authors attribute to possible other compensatory mechanisms). Although the effect of ablating *Gaq/11* signaling on endfeet size was shown in the presence of taltirelin, it was not shown in the absence of taltirelin (Figure 4). Although I recognize the experimental difficulty of many of these experiments, and I acknowledge that the results presented in the figures represent good support for the authors' overall model, additional experiments that extend and link these results to the overall physiological/HPT context would further strengthen the authors' overall conclusions. At minimum the authors should discuss these issues at greater length; in particular, I feel that the legitimacy of using the hM3D/CNO chemogenetic system as an accurate mimic of physiological TRH signaling should be more clearly justified.

2. The experiments were analyzed employing a variety of different statistical tests (e.g. the data in Figure 2f and Figure 4 were analyzed using a two-tailed Mann-Whitney U-test, the data in Figure 1c were analyzed using a Kruskal-Wallis with post-hoc Dunn's test, and the data in Figure 3 were analyzed using a Student's t-test). Given that a number of the observed effects are relatively modest in magnitude and statistically close to the $p \leq 0.05$ often considered (if arbitrarily) the acceptable "standard" it would be helpful if the authors expanded on their description of these analyses in the Materials and Methods to more fully explain their rationale for applying these different specific statistical tests to these different experimental contexts.

3. Although β 2-tanycytes are the most relevant for the phenomena under study here, the experiments in Figure 2 utilize a *GlastCreER/hM3D-Cherry* system to generate cell-specific chemogenetic induction of G protein signaling and visualization of endfeet morphology. *Glast*-driven expression has been reported to be limited to α -tanycytes with little or no expression in β -tanycytes. (Ronins et al. *Nature Communications* 4, article 2049 (2013) doi: 10.1038/incomms3049). The authors note this in their Discussion but state that to the contrary, *Glast* is expressed in all subtypes of tanycytes and that they were able to recombine the Flex system so as to also express hM3D-mCherry in β -tanycytes in this genotype (lines 258-260). Given the relevance of cell specificity to the interpretation of these experiments, the authors should further discuss this issue, describe the modification of the Flex system that permitted use of this system in β -tanycytes, and more explicitly document the success of their strategy.

4. ATP was used as a control in the heat maps in Figure 1d; it would strengthen this figure if the ATP response was quantified for multiple experiments and the data incorporated into Figure 1c. Similarly the authors provide a representative trace of the response of endfeet diameter to CNO in control (untransduced) and in Tan3D mice (Figure 2c), but quantify the results only for the Tan3D group (-/+ CNO, Figure 2f); the control data should also be quantified and included in Figure 2f. Data obtained in the absence of taltirelin in Figure 4 and in Supplemental Figure 5b should also be provided; these in particular would help confirm that TRHR agonists themselves, not just the artificial chemogenetic hM3D approach, can modulate TSH levels in a *Gaq/11* dependent manner.

5. Both the anatomical structures involved and the experimental procedures are relatively complex and likely make reading and understanding the manuscript difficult for the general reader. Perhaps adding a table summarizing the key features of the various mouse genetic backgrounds and the corresponding reporter gene/CRE constructs would be helpful (including, where relevant, the tissue specificities previously established for each of these systems, including cell types other than the tanycytes analyzed in the current study). Further, incorporating a sketch of the overall anatomy of the ME as Figure 1 (such as the one in the current Supplemental Figure 3) would also be a useful aid to the non-expert reader, allowing them to both better follow the introduction/materials and methods and to better understand the rationale of the experiments as presented in the Results section.

We thank the reviewers for their helpful comments. In response to the reviewers' comments we have performed further experiments and have revised the manuscript as outlined below. The revised manuscript is clearly improved in our opinion. Changes in the manuscript have been marked by red letters.

Reviewer #1 (Remarks to the Author):

The manuscript by Muller-Fielitz and colleagues explores novel regulatory processes controlling neurosecretion operating at the tanycytic endfeet of the gliovascular unit in the median eminence, a neurohemal organ located in the ventral part of the tuberal region of the hypothalamus. In particular, using cutting-edge approaches and original virogenic tools, the authors provide the first in vivo demonstration that tanycytes control the bioavailability of the thyroid-releasing hormone (TRH) and its release into the pituitary portal blood vessels, when TRH neurons are activated. This is a conceptually important and logical manuscript. Technically, it is a "tour de force" compilation of many advanced techniques, which clearly increase its appeal. These methods are sound and sufficient to support the overarching hypothesis. The paper certainly is an important piece of work, which would feature well in Nature Communications.

Response: We are grateful for the positive evaluation by the reviewer.

The manuscript reads beautifully and has been carefully prepared. My specific comments are elaborated bellow:

1. The authors show that by chemogenetically activating G-alpha-q/11 in tanycytes using the DREADD technology in a GlastCreERT2 mouse background, they change the morphology of tanycyte endfeet within one hour after CNO administration. The efficiency of the tamoxifen-induced cre-mediated gene recombination being highly variable from one animal to another, how did the authors control the fact that the changes of the expansion of tanycytic endfeet they see in Figure 2d-f was actually due to the effect of CNO? It would be more convincing if the authors could complement these data by experiments using living brain slices in which mCherry-positive tanycytic endfeet could be imaged before and after CNO application into the bath (i.e., like they did in Figure 1 and 2C). An alternative, would be to show in Figure 2d and 2e a lower magnification image illustrating the fact that the same number of tanycytes have been transduced in each condition (i.e., number of mCherry-positive tanycytic cell bodies lining the wall of the third ventricle).

Response: The reviewer raises the important point of variability in the tamoxifen-induced Cre-mediated recombination. To control for chance variation in the recombination efficiency, we repeated the experiments and included the new data in Fig. 3 of the revised manuscript (former Fig. 2, sample size now 6 animals per group). These experiments confirmed the conclusion that $G\alpha_{q/11}$ signaling increases the endfoot size of tanycytes. In addition, we used a tamoxifen-independent approach. By injecting two AAV vectors (AAV-Dio2-iCre-2A-GFP and AAV-CAG-flex(hM3D-mCherry)) we were able to label tanycyte endfeet and to activate $G\alpha_{q/11}$ signaling. Also in these tamoxifen-independent experiments, treatment with CNO or with the TRH agonist taltirelin increased the endfoot size in wild-type tanycytes but taltirelin had no effect in $G\alpha_{q/11}$ -deficient cells (Fig. 5, lines 178-186). For the analysis of the endfoot size we followed a strict blinding strategy with the

experimenter not knowing the treatment or the genotype of samples (lines 376-377). Thus, we are very confident that $G\alpha_{q/11}$ signaling influences endfoot size.

Nevertheless, we agree that experiments in living brain slices, as suggested by the reviewer, could be very informative and would allow further analysis of the mechanisms underlying endfoot expansion. So far, we have performed pilot experiments to establish this technique. However, we had problems in controlling movements of the slices to perform repetitive high-resolution confocal imaging of tancyte endfeet. In the future, we will try hard to set up this technology.

As suggested by the reviewer we have included low magnification overviews to show that the same number of tancytes have been transduced (Supplementary Fig. 3).

2. The authors convincingly demonstrate that manipulating tancyte function tightly modulates the activity of the hypothalamic-pituitary-thyroid hormone axis when TRH neurons are active, as measured by TSH release and activity. Are there any physiological and pathological states in which manipulating tancytic function could alter the adaptive response of the thyroid axis and result in measurable changes in metabolism (e.g., changes in temperature)? This point can be touched upon in the discussion.

Response: We highly appreciate this important comment. To investigate the physiological relevance of the tancytic regulation of the HPT axis we determined the expression of T3-dependent genes in the liver. When the tancytic brake was released, activation of TRH neurons induced the T3-dependent genes *Thrsp* and *Fasn* in the liver (Fig. 7h and I, lines 234-237) suggesting a profound effect on metabolism.

As typical pathological states, in which tancyte function could alter the adaptive response of the thyroid axis, we turned to hypo- and hyperthyroidism. When modulating the HPT axis we found that the effect of TRH receptor activation on tancytic $[Ca^{2+}]_i$ was reduced in hypothyroid mice releasing the tancytic brake on the HPT axis. In contrast, under hyperthyroid conditions the response of $[Ca^{2+}]_i$ to TRH receptor activation was increased, which will reduce TRH secretion from the hypothalamus. As a full description of the experiments might exceed the scope of the current manuscript, we prefer to mention these findings as “data not shown” and to discuss the implications for pathological states as suggested by the reviewer (lines 331-338).

Minors:

i) Line 317, what is the serotype of the AVV used?

Response: We had mentioned the serotype of the AAVs at the beginning of the Results section and in the Method section “Recombinant AAV production”. However, we agree with the reviewer that this information is also expected in the Methods section “Mice” and have revised the manuscript accordingly (line 362-363).

ii) Line 324, at what age were the animals injected with tamoxifen? If different ages have been used, was there any changes in treatment efficiency between ages?

Response: The animals were 8-10 weeks of age when they were injected with tamoxifen (line 369).

Reviewer #2 (Remarks to the Author):

Tanycytes are found in close apposition to the endings of TRH neurons in the median eminence raising the possibility that they may play an important role in controlling the output of the hypothalamic-pituitary-thyroid axis (HPT). The role of tanycytes in regulating the release of GnRH has been investigated, but to the best of my knowledge, this is the first exploration for a role in TRH release.

The authors use a number of different approaches to show that activation of TRH1 receptors on tanycytes modulates HPT output. They propose a model whereby tanycytes inhibit the HPT. This requires activation of TRH1 receptors on tanycytes which increases Ca, causing an increase in the size of the endfeet

Each of the observations taken independently, is interesting. I do, however, have concerns about the use of the tools. In particular, I'm puzzled by the multiple mouse lines and am not convinced about the specificity of the targeting. In addition, since the mechanistic experiments are conducted in vitro, I would like to see a higher level of sophistication and resolution for a journal like Nature Communications. In particular, I think single-cell resolution for Ca changes should be required. I also think the data, as presented, for the endfoot volume changes makes it difficult for readers to discern the changes for themselves.

Response: We accept the point that the multiple mouse models that we have used in this study may be puzzling for readers. However, we feel that the description of two strategies of targeting tanycytes is a strength of our study and could be helpful for other work in the field. To increase readability of the text we have therefore included a table that summarizes key features of the mouse models (Supplementary Table 1). Furthermore, we tried to better explain differences between the two main approaches (lines 280-289).

To achieve single cell resolution in $[Ca^{2+}]_i$ imaging we have used confocal microscopy and included the data in Fig. 1 of the revised manuscript.

Concerning the targeting of tanycytes we are very positive that the two strategies that we have developed are selective as described below.

Major Concerns

1) I find the use of multiple mouse lines confusing. For example, the initial examination of TRHR1-dependent increases in endfoot size is done in *GlastCreERT2* mice. Then, the authors show that the change in endfoot size induced by taltirenin is eliminated by genetic elimination of *Gq11a* in *Tan3D* mice. Why the different strains? Unless I have missed something, I do not see the need for using the *GlastCreERT2* mice.

Response: We apologize that the use of multiple mouse lines seems to be confusing. The reviewer is right that there is no absolute need in our study to use the *GlastCreER^{T2}* line in addition to the AAV-Dio2-iCre-2A-GFP targeting of tanycytes. Nevertheless, we opt to keep the data in the manuscript because it represents the first description that this mouse line, in combination with the intraventricular injection of AAV-based vectors, can be used to selectively manipulate tanycytes *in vivo*. With this approach recombination is inducible by tamoxifen, a feature that might be handy for future work. In the case of our study, the use of *GlastCreER^{T2}* mice has provided an independent line of evidence that targeting $G\alpha_{q/11}$ signaling in tanycytes is able to modulate the size of their endfeet. In order to avoid any confusion, we elaborate in the revised manuscript that we use two strategies to

target tanycytes *in vivo* and discuss their pros and cons (lines 280-289).

2) Additionally, I am concerned about potential contributions from astrocytes in the *GlastERT2*. The authors are aware of this as they show images in supplementary data that are consistent with little/no expression in astrocyte. While this makes a stronger case for tanycytes, I'm puzzled that there is no overlap with astrocytes.

Response: The reviewer is right that *GlastCreER^{T2}* mice show inducible recombination in tanycytes and in astrocytes. However, in our study the Cre-dependent target gene was supplied by the intraventricular injection of the vector AAV-CAG-flex(hM3D-mCherry) which is trapped in the ventricle wall. In contrast to tanycytes, astrocytes do not extend to the ventricle wall explaining why the combination of *GlastCreER^{T2}* plus intraventricular AAV vectors results in tanycyte selectivity. As we realize that this strategy was not sufficiently explained in the original manuscript, we have provided more details in the revised manuscript (lines 124-138, lines 149-151, lines 280-289). In addition to the close-ups of the median eminence provided in Fig. 3a-d, we include overviews of brain sections of *GlastCreER^{T2}* mice injected with AAV-CAG-flex(hM3D-mCherry) that illustrate the specificity of the approach (Supplementary Fig. 3).

3) The changes in Ca in tanycytes shown in figure 1 is a good starting point. A cellular-level resolution of the Ca signal would be useful here. Also, it's not clear how the data were quantified. Is it better to examine the peak or integrate the Ca signal? Single-cell resolution, 2-photon imaging would add substantially to these observations.

Response: Appreciating this comment we have performed confocal imaging of tanycytes. With this technology we have imaged $[Ca^{2+}]_i$ at single cell resolution (Fig. 1c and d, lines 94-97, lines 448-452). Due to the additional data, we split the former Fig. 1 into two figures (Fig. 1, Fig. 2). Concerning the quantification, we apologize for not providing all information in the original version of the manuscript. We used maximal $[Ca^{2+}]_i$ signals, as specified now in the legend to Fig. 2a,b and d and in the "Methods" section (lines 470-471).

4) The CNO experiment (n=4) is underpowered. I would feel more confident with additional experiments to solidify this observation.

Response: Following this comment we have performed additional experiments to solidify the observation in Fig. 3 (former Fig. 2). This important finding is also strongly supported by the additional and independent set of experiments reported in Fig. 5 (described in lines 178-186). Thus, we are very confident that this is a robust result.

5) The supplementary figures look beautiful, but there is no quantification. It is imperative that there is selective targeting of tanycytes with the construct. Whilst this appears to be the case, the reader cannot be certain without quantification.

Response: We appreciate this comment and have quantified cells expressing hM3D-mCherry as a sign of recombination in various brain areas. Cell counts are provided in the revised manuscript (Supplementary Fig. 4f, g; lines 161-163). The quantification revealed that recombination occurred mostly in tanycytes. As expected, there was some variability in the rate of off-target recombination between vector preps. Notably, the quantification shown in Supplementary Fig. 4f, g was performed with the prep showing the highest rate of off-target recombination in our study. To provide more evidence of the selective targeting we included an overview image in the revised

manuscript showing recombination in *GlastCreER^{T2}* mice injected with AAV-CAG-flex(hM3D-mCherry) (Supplementary Fig. 3).

Minor

Why were plasma TSH concentrations measured only 2 hours after stimulation of TRH cells? A better temporal quantification of changes in TSH would be helpful.

Response: Frequent blood sampling for TSH quantification could affect its plasma concentrations. Therefore, we had to use separate experimental cohorts for each time point. Guided by published studies that used TRH or analogues we selected two time points, i.e. 45 min and 2 hours, after CNO stimulation. The data were already present in the original manuscript but not clearly labeled. In the revised manuscript, we have now highlighted the two time points in Fig. 6.

I'm concerned by the mcherry expression in TRH neurons in Fig 5C.

Response: We apologize that the mCherry expression in TRH neurons was not clear in the original figure (revised Fig. 6c). This was due to the inclusion of DAPI staining that really did not add very much to the figure. We have now omitted DAPI staining and believe that the revised figure is much improved.

The abstract states that tanycytes are located in close proximity to TRH neurons. This is not accurate. The neurons (cell bodies) are found in the PVN and the axon terminals of these neurons are in the median eminence, adjacent to tanycytes.

Response: We thank the reviewer for drawing our attention to the imprecise wording. We meant to say that TRH nerve endings are close to tanycytes and have corrected the abstract accordingly (lines 29-30).

Reviewer #3 (Remarks to the Author):

The hypothalamic-pituitary-thyroid (HPT) axis is normally tightly regulated so as to maintain thyroid hormone levels at the correct physiological concentrations. This manuscript reports a new potential role for hypothalamic β 2-tanycytes in influencing this homeostatic regulation of the HPT axis. The authors report that thyrotropin-releasing hormone (TRH), operating through a TRH receptor-1/ $G\alpha q/11$ mediated mechanism in the β -tanycytes in the median eminence, produces increases in the size of the tanycyte endfeet that project onto the pituitary vessels and increases in the activity of the TRH-degrading ectoenzyme TRH-DE. These events would be expected to impair thyroid stimulating hormone (TSH) release by the TRH-responsive pituitary cells in the median eminence through physical shielding and/or by increased TRH degradation. Consistent with this prediction, the authors report that blocking TRH signaling in tanycytes by ablating $G\alpha q/11$ expression in these cells enhances the response of the HPT axis to the chemogenetic activation of TRH neurons. The authors conclude that their study identifies "new TRH- and $G\alpha q11$ -dependent mechanisms in the median eminence by which tanycytes control the activity of the HPT axis."

Overall this is a fairly interesting study. The identification of a possible role of tanycytes in modulating TRH availability is itself intriguing and would not only reveal an unanticipated regulation of the HPT axis but would also support the more general concept that tanycytes can function as modulators of a variety of endocrine signaling pathways. The authors convincingly establish that TRH/taltirelin can alter Gαq/11-calcium signaling in β2-tanycytes, that chemogenetic stimulation of Gαq/11 pathways in these tanycyte cells (by expression of ectopic hM3D and CNO treatment) results in moderate increases in the diameters of the endfeet of these cells in a Gαq/11 dependent fashion, and that chemogenetic stimulation of TRH release by the ME-associated TRH neurons produces enhanced serum levels of TSH which are further increased when Gαq/11 signaling was ablated in the tanycytes. Whereas taltirelin (a TRHR agonist) increased TRH-DE enzymatic levels in control mice, ablating Gαq/11 signaling in the tanycytes resulted in decreased levels of TRH-DE enzymatic activity and in smaller endfeet (the latter when compared to control mice with both assayed in the presence of taltirelin).

These individual observations are valuable and are consistent with the authors' overall conclusion that β-tanycytes, in response to TRH, can alter/regulate the availability of this hormone in the ME. However some question still remains if these individual experimental observations add together sufficiently to completely support the authors' overall conclusion, and if they do, how substantial a contribution these phenomena actually represent in the physiological regulation of the HPT axis. Further, much of the analysis in this manuscript relies heavily on the use of fairly complex combinations of genetically manipulated mice, the introduction of a variety of ectopic CRE and reporter constructs into anatomically specific regions of the brain, and the use of non-physiological chemical inducers and inhibitors of specific pathways. The technical complexity of these methodologies raises some issues as to the biological relevance of the results obtained, and although the manuscript is well written, also makes it somewhat difficult for an average reader to easily follow all the logic and implications of the experiments. The authors should address these issues in more detail. Specific concerns and suggestions are detailed below:

Response: We agree with the reviewer that the physiological relevance is of utmost importance. The HPT axis is regulated at multiple levels, i.e., the paraventricular nucleus, the pituitary gland, the thyroid, and, finally as we show here, in the median eminence. The redundant control of the axis prevents fatal consequences if one of these mechanisms fails. Therefore, it is expected that interfering with tanycyte function is compensated at other levels if the overall activity of the axis is investigated. To focus on the TRH release from the hypothalamus we had to acutely stimulate TRH neurons (justified in lines 267-270). Under these conditions, tanycyte function for the HPT axis proved to be important and to have consequences for peripheral organs such as the liver, as we show in the revised manuscript (Fig. 7h and i, lines 234-237). Thus, our data suggest that tanycytes contribute to the stability of the axis and could play an important role in controlling the speed in which changes of the HPT axis occur. Although it is very interesting, we only shortly address these aspects in order to be not too speculative in the discussion (lines 331-338).

1. As noted above, some of the evidence provided that tanycytes regulate the HPT axis is indirect. For example, although acute stimulation through the CNO/hM3D approach detectably altered serum TSH levels and this was further enhanced by ablating Gαq/11 signaling in the tanycytes (Figure 6), acute treatment with TRH itself (or a similar agonist) increased TRH-DE levels without a reported effect on TSH plasma levels; also, abolishing Gαq/11 signaling in these cells resulted in no detectable change in steady state serum T3, T4, or TSH levels despite reducing the associated TRH-

DE levels (Figure 3, the lack of which the authors attribute to possible other compensatory mechanisms). Although the effect of ablating $G\alpha_{q/11}$ signaling on endfeet size was shown in the presence of taltirelin, it was not shown in the absence of taltirelin (Figure 4). Although I recognize the experimental difficulty of many of these experiments, and I acknowledge that the results presented in the figures represent good support for the authors' overall model, additional experiments that extend and link these results to the overall physiological/HPT context would further strengthen the authors' overall conclusions. At minimum the authors should discuss these issues at greater length; in particular, I feel that the legitimacy of using the hM3D/CNO chemogenetic system as an accurate mimic of physiological TRH signaling should be more clearly justified.

Response: We are thankful for raising these important issues. The fact that TRH itself or taltirelin, a TRH agonist, had no effect on TSH plasma levels after ablating $G\alpha_{q/11}$ signaling in tanycytes is expected, because exogenous TRH or its agonist stimulate TSH secretion in the pituitary, independent of the release of endogenous TRH from the hypothalamus. To investigate hypothalamic secretion of endogenous TRH we had to stimulate TRH neurons. For the first time this task could be accomplished by using a chemogenetic approach. Following the reviewer's advice we try to more clearly justify the legitimacy of this approach in the revised manuscript (lines 267-270). Our data suggest that tanycytic $G\alpha_{q/11}$ signaling selectively modulates TRH release from the hypothalamus as evidenced by the stimulation of TSH secretion.

The reviewer is right to note that we had not investigated the effect of ablating $G\alpha_{q/11}$ signaling on endfoot size in the absence of taltirelin. To correct this shortcoming we have repeated the experiment including these groups (Fig. 5). The results have clearly confirmed that the effect of TRH neuron activation or of exogenous taltirelin on endfoot size depends on $G\alpha_{q/11}$ signaling in tanycytes.

2. The experiments were analyzed employing a variety of different statistical tests (e.g. the data in Figure 2f and Figure 4 were analyzed using a two-tailed Mann-Whitney U-test, the data in Figure 1c were analyzed using a Kruskal-Wallis with post-hoc Dunn's test, and the data in Figure 3 were analyzed using a Student's t-test). Given that a number of the observed effects are relatively modest in magnitude and statistically close to the $p \leq 0.05$ often considered (if arbitrarily) the acceptable "standard" it would be helpful if the authors expanded on their description of these analyses in the Materials and Methods to more fully explain their rationale for applying these different specific statistical tests to these different experimental contexts.

Response: We appreciate the comment and have better justified the use of statistical tests (lines 599-606). In the original manuscript we had analyzed the data in the former Fig. 2f and Fig. 4 with the Mann-Whitney U-test because the sample size was below 5. As the sample size in both figures was increased by additional experiments during the revision, we are now able to use Student's t-test (Fig. 3g) and one-way ANOVA (Fig. 5a). Now, the Mann-Whitney U-test is no longer used.

We used the Kruskal-Wallis test as a non-parametric test in Fig. 2a and Fig. 2d (former Fig. 1c), because the variance differed between groups. After ANOVA, we used the Bonferroni test as one of the most stringent post-hoc tests to ensure the validity of our conclusions.

3. Although β 2-tanycytes are the most relevant for the phenomena under study here, the experiments in Figure 2 utilize a $G\text{lastCreER/hM3D-Cherry}$ system to generate cell-specific chemogenetic induction of G protein signaling and visualization of endfeet morphology. $G\text{last}$ -driven expression has been reported to be limited to α -tanycytes with little or no expression in β -tanycytes. (Ronins et al. Nature Communications 4, article 2049 (2013) doi:

10.1038/incomms3049). The authors note this in their Discussion but state that to the contrary, *Glast* is expressed in all subtypes of tanycytes and that they were able to recombine the Flex system so as to also express hM3D-mCherry in β -tanycytes in this genotype (lines 258-260). Given the relevance of cell specificity to the interpretation of these experiments, the authors should further discuss this issue, describe the modification of the Flex system that permitted use of this system in β -tanycytes, and more explicitly document the success of their strategy.

Response: We are aware that recombination in β -tanycytes of *GlastCreER^{T2}* contradicts the study of Ronins et al. (2013) that reported recombination only in α -tanycytes. Our conclusion that there is additional recombination in β -tanycytes is based on the expression of mCherry in β -tanycytes that are characterized by their distinct localization at the bottom of the 3rd ventricle and cannot be mixed up with α -tanycytes. Also leakiness of the reporter can be excluded because we found no signal when injecting vector from the same prep in wild-type mice. Probably, the discrepancy is due to the highly sensitive detection of Cre activity with the flex system that we used. The information that there is recombination in β -tanycytes of the *GlastCreER^{T2}* mouse line might be relevant for future studies. Importantly, we confirmed the expansion of β -tanycyte endfeet with an independent approach using the AAV-Dio2-iCre-2A-GFP and the AAV-CAG-flex(hM3D-mCherry) vectors (Fig. 5, lines 178-186). Also, when investigating modulation of the HPT axis by measuring hormone levels we did not employ the *GlastCreER^{T2}* line but used the vector AAV-Dio2-iCre-2A-GFP (Fig. 7). Therefore, our conclusions about the role of $G\alpha_{q/11}$ signaling in tanycytes do not exclusively rely on the specificity of the *GlastCreER^{T2}* mouse line.

4. ATP was used as a control in the heat maps in Figure 1d; it would strengthen this figure if the ATP response was quantified for multiple experiments and the data incorporated into Figure 1c.

Response: Appreciating the comment on ATP we have quantified the response to ATP and have included the data in Fig. 2b of the revised manuscript (lines 99-100).

Similarly the authors provide a representative trace of the response of endfeet diameter to CNO in control (untransduced) and in Tan3D mice (Figure 2c) , but quantify the results only for the Tan3D group (-/+ CNO, Figure 2f); the control data should also be quantified and included in Figure 2f.

Response: Concerning the data depicted in Fig. 3f and Fig. 3g (former Fig. 2c and Fig. 2f), there seems to be a misunderstanding. In Fig. 3f we show the effect of CNO treatment on $[Ca^{2+}]_i$ to verify that the chemogenetic stimulation worked. We apologize for not describing the figure clearly. In the revised version the figure legend clearly describes the nature of the experiment. In non-transduced control mice tanycyte endfeet would not be labeled.

Data obtained in the absence of taltirelin in Figure 4 and in Supplementary Figure 5b should also be provided; these in particular would help confirm that TRHR agonists themselves, not just the artificial chemogenetic hM3D approach, can modulate TSH levels in a $G\alpha_{q/11}$ dependent manner.

Response: As suggested by the reviewer, we have investigated endfoot size also in the absence of taltirelin and included the data in Fig. 5 (former Fig. 4; described in lines 178-186). The data confirm that taltirelin is able to increase the endfoot size of tanycytes. Furthermore, we have included TSH levels in mice that were not treated with taltirelin in Supplementary Fig. 5b.

5. Both the anatomical structures involved and the experimental procedures are relatively complex and likely make reading and understanding the manuscript difficult for the general reader. Perhaps

adding a table summarizing the key features of the various mouse genetic backgrounds and the corresponding reporter gene/CRE constructs would be helpful (including, where relevant, the tissue specificities previously established for each of these systems, including cell types other than the tanycytes analyzed in the current study). Further, incorporating a sketch of the overall anatomy of the ME as Figure 1 (such as the one in the current Supplementary Figure 3) would also be a useful aid to the non-expert reader, allowing them to both better follow the introduction/materials and methods and to better understand the rationale of the experiments as presented in the Results section.

Response: We highly appreciate this comment. To help readers we followed the reviewer's suggestion and included a table (Supplementary Table 1; reference in the main text, lines 111-112) to summarize key features of the various mouse models. In addition, we shifted the former Supplementary Fig. 3 into Fig. 3 of the revised manuscript to provide an overview about the anatomy of the ME.

Tissue specificity of our mouse models should be no issue because we locally administered Cre expressing vectors into the lateral ventricle for analyzing systemic parameters. The *GlastCreER*^{T2} driver line was only employed to investigate tanycyte morphology or $[Ca^{2+}]_i$.

Supplementary Table 1: Summary of the mouse models generated in this study. LV, lateral ventricle; PVN, paraventricular nucleus; C57Bl/6, wild-type strain.

	AAV (site of injection)	Genotype of mice	Short name of mice	Purpose/Effect
Supplementary Fig. 1	AAV-CAG-Cre (LV)	Ai14 Cre reporter	Ai14	Expression of tdTomato in tanycytes
Fig. 1	AAV-CAG-GCamP6s (LV)	C57Bl/6	Bl6	Expression of GCamP6s in tanycytes
Fig. 2	AAV-CAG-GCamP6s (LV)	C57Bl/6	Bl6	Expression of GCamP6s in tanycytes of C57Bl/6 mice
		Trhr1 ^{-/-}	Trhr1 ^{-/-}	Expression of GCamP6s in tanycytes of Trhr1 ^{-/-} mice
		Trhr2 ^{-/-}	Trhr2 ^{-/-}	Expression of GCamP6s in tanycytes of mice Trhr2 ^{-/-} mice
		GlastCreER ^{T2} ; Gα_q ^{fl/fl} :: Gα₁₁ ^{-/-}	Gα_{q11} ^{gliaKO}	Expression of GCamP6s in tanycytes of mice with deficiency of Gα_{q11} proteins in glia and tanycytes
Fig. 3, Supplementary Fig. 3	AAV-CAG-flex(hM3D-mCherry) (LV)	GlastCreER ^{T2}	gTan ^{3D}	Selective expression of hM3D-mCherry in tanycytes
Fig. 4a, b, Supplementary Fig. 4a-e	AAV-Dio2-iCre-2A-GFP (LV)	Ai14 Cre reporter	Ai14	Selective expression of tdTomato in tanycytes
Fig. 4c-g, Supplementary Fig. 5	AAV-Dio2-GFP (LV)	Gα_q ^{fl/fl} :: Gα₁₁ ^{-/+}	Gα_{q11} ^{Con}	Littermate control for Gα_{q11} ^{tanKO}
	AAV-Dio2-iCre-2A-GFP (LV)	Gα_q ^{fl/fl} :: Gα₁₁ ^{-/-}	Gα_{q11} ^{tanKO}	Selective deletion of Gα_{q11} proteins in tanycytes
Fig. 5, Supplementary Fig. 4f and g	AAV-Dio2-iCre-2A-GFP (LV) + AAV-CAG-flex(hM3D-mCherry) (LV)	C57Bl/6	dTan ^{3D} -Bl6	Tanycyte selective expression of hM3D-mCherry in C57Bl/6 mice
		Gα_q ^{fl/fl} :: Gα₁₁ ^{-/-}	dTan ^{3D} - Gα_{q11} ^{tanKO}	Tanycyte selective expression of hM3D-mCherry and deletion of Gα_{q11}
Fig. 6, Supplementary Fig. 6	-	C57Bl/6	PVN ^{Con} -Bl6	Chemogenetic stimulation of TRH neurons in C57Bl/6 mice
	AAV-TRH-hM3D-mCherry (PVN)		PVN ^{3D} -Bl6	
	-	Trhr1 ^{-/-}	PVN ^{Con} - Trhr1 ^{-/-}	Chemogenetic stimulation of TRH neurons in Trhr1 ^{-/-} mice
	AAV-TRH-hM3D-mCherry (PVN)		PVN ^{3D} - Trhr1 ^{-/-}	
Fig. 7	AAV-Dio2-GFP (LV)	Gα_q ^{fl/fl} :: Gα₁₁ ^{-/+}	PVN ^{Con} - Gα_{q11} ^{Con}	Controls for PVN ^{3D} - Gα_{q11} ^{Con}
	AAV-Dio2-GFP (LV) + AAV-TRH-hM3D-mCherry (PVN)	Gα_q ^{fl/fl} :: Gα₁₁ ^{-/+}	PVN ^{3D} - Gα_{q11} ^{Con}	Chemogenetic stimulation of TRH neurons in control mice
	AAV-Dio2-iCre-2A-GFP (LV) + AAV-TRH-hM3D-mCherry (PVN)	Gα_q ^{fl/fl} :: Gα₁₁ ^{-/-}	PVN ^{3D} - Gα_{q11} ^{tanKO}	Chemogenetic stimulation of TRH neurons plus selective deletion of Gα_{q11} proteins in tanycytes

REVIEWERS' COMMENTS:

Reviewer #1 (Remarks to the Author):

The authors satisfactorily answered all reviewer's comments.

Reviewer #2 (Remarks to the Author):

The authors have done a nice job of addressing most of my concerns. The revised version is vastly improved from the original submission and for that, the authors should be commended.

I do, however, have a few remaining issues.

I raised a concern about mCherry expression in TRH neurons. Ostensibly, this denotes cells that express Hm3D. If the targeting is accurate, then only TRH cells should be mCherry positive. The authors write in the results section, "we injected the vector AAV-TRH-hM3D-mCherry, in which the expression of the fusion protein hM3D-mCherry was under transcriptional control of the rat TRH promoter²⁹, bilaterally into the PVN of C57Bl/6 mice (PVN3D-BI6). After 2 weeks hM3D- mCherry could be detected in Trh mRNA expressing neurons of the PVN." If this truly is hM3D-mCherry expression under transcriptional control of the TRH promoter, then why, in fig. 6, is there mCherry expression in cells that do not express TRH mRNA.

I also think the title does not accurately represent the key findings in the manuscript. The authors are not examining, 'activity' per se. This term is usually used to denote neuronal spiking. I would be more comfortable with the term, 'output.' In other words, the title should be, "Tanycytes control the activity of the hypothalamic-pituitary-thyroid axis."

Reviewer #3 (Remarks to the Author):

The revised manuscript adequately addresses my prior concerns.

We would like to thank reviewer #2 for his/her thorough comments on our manuscript.

The authors have done a nice job of addressing most of my concerns. The revised version is vastly improved from the original submission and for that, the authors should be commended.

Response: We are glad about this overall positive evaluation.

I do, however, have a few remaining issues.

I raised a concern about mCherry expression in TRH neurons. Ostensibly, this denotes cells that express Hm3D. If the targeting is accurate, then only TRH cells should be mCherry positive. The authors write in the results section, “we injected the vector AAV-TRH-hM3D-mCherry, in which the expression of the fusion protein hM3D-mCherry was under transcriptional control of the rat TRH promoter²⁹, bilaterally into the PVN of C57Bl/6 mice (PVN3D-BI6). After 2 weeks hM3D- mCherry could be detected in *Trh* mRNA expressing neurons of the PVN.” If this truly is hM3D-mCherry expression under transcriptional control of the TRH promoter, then why, in fig. 6, is there mCherry expression in cells that do not express TRH mRNA.

Response: The reviewer is right that hM3D-mCherry expression was found in cells expressing *Trh* mRNA but also in cells that were negative for *Trh* mRNA. This finding has been explicitly stated in the text and has been highlighted in Fig. 6 by marking the cells with red and yellow arrowheads. There are two possible explanations for the presence of *Trh* mRNA negative cells that express hM3D-mCherry. First, it is possible that the rat promoter that we have used, extending from – 776 to +84, did not convey absolutely cell specific gene transcription. Because of the limited capacity of AAV vectors we were not able to use the full locus. It is well known that generally cell specificity increases with the length of regulatory sequences that control gene expression. Nevertheless, we are confident that the promoter fragment from -776 to +84 conveys a high degree of cell specific regulation, because this sequence or even shorter fragments of it have been used in several previous studies *in vivo* (Balkan et al., 1998; Guissouma et al., 1998; Kouidhi et al., 2010) and were found to closely reflected the regulation of the endogenous *Trh* gene. Moreover, we performed an extensive functional and morphological characterization of hM3D-mCherry expression (Fig. 6). Importantly, the hM3D-mCherry positive nerve endings in the median eminence could be clearly dissociated from vasopressin, oxytocin or GnRH positive cells. Finally, we did not elicit an increase of ACTH at a time when we saw elevated TRH levels 120 min after CNO administration. All these observations argue against the assumption that the *Trh* promoter fragment that we have used has limited specificity and suggest to us as an alternative second explanation that the *in situ* hybridization of *Trh* mRNA may be not sensitive enough to detect all TRH neurons. In order to combine the *in situ* hybridization of *Trh* mRNA and with the immunohistochemical detection of hM3D-mCherry, we performed *in situ* hybridization with a digoxigenin-labeled probe. This form of labeling has the disadvantage that it is less sensitive than radioactive labeling of probes. The failure to detect low *Trh* mRNA expression in some of the TRH neurons may very well explain the finding of *Trh* mRNA negative cells that express hM3D-mCherry. We mention this option in the revised manuscript (lines 201-203).

I also think the title does not accurately represent the key findings in the manuscript. The authors are not examining, ‘activity’ per se. This term is usually used to denote neuronal spiking. I would be more comfortable with the term, ‘output.’ In other words, the title should be, “Tanycytes control the activity of the hypothalamic-pituitary-thyroid axis.”

Response: We appreciate the reviewer’s support but disagree with the view that ‘activity’ is only

used for neuronal spiking. Numerous articles in the scientific literature use the term 'activity' in the context of a hypothalamic-pituitary axis. A prominent example is the paper by Geibel et al. (2014) in Nature Communications that states in the abstract that 'Glucocorticoids control HPA axis activity through negative feedback to the pituitary gland and the central nervous system (CNS)' (Geibel et al., 2014). Therefore, we would prefer not to change the title but leave the decision to the editor's discretion.

References

- Balkan, W., Tavianini, M.A., Gkonos, P.J., and Roos, B.A. (1998). Expression of rat thyrotropin-releasing hormone (TRH) gene in TRH-producing tissues of transgenic mice requires sequences located in exon 1. *Endocrinology* *139*, 252-259.
- Geibel, M., Badurek, S., Horn, J.M., Vatanashevanopakorn, C., Koudelka, J., Wunderlich, C.M., Bronneke, H.S., Wunderlich, F.T., and Minichiello, L. (2014). Ablation of TrkB signalling in CCK neurons results in hypercortisolism and obesity. *Nat Commun* *5*, 3427.
- Guissouma, H., Ghorbel, M.T., Seugnet, I., Ouatas, T., and Demeneix, B.A. (1998). Physiological regulation of hypothalamic TRH transcription in vivo is T3 receptor isoform specific. *Faseb j* *12*, 1755-1764.
- Kouidhi, S., Seugnet, I., Decherf, S., Guissouma, H., Elgaaied, A.B., Demeneix, B., and Clerget-Froidevaux, M.S. (2010). Peroxisome proliferator-activated receptor-gamma (PPARgamma) modulates hypothalamic Trh regulation in vivo. *Mol Cell Endocrinol* *317*, 44-52.